# Neuronal DSCAM regulates the peri-synaptic localization of GLAST in Bergmann glia for functional synapse formation

Ken-ichi Dewa[1,2,3], Nariko Arimura [1,4] ✉, Wataru Kakegawa[5], Masayuki Itoh[5], Toma Adachi[1], Satoshi Miyashita[1,6], Yukiko U. Inoue [1], Kento Hizawa[4], Kei Hori[1], Natsumi Honjoya[4], Haruya Yagishita[4], Shinichiro Taya[1,7], Taisuke Miyazaki[8], Chika Usui[9], Shoji Tatsumoto[9], Akiko Tsuzuki[1], Hirotomo Uetake[1,10], Kazuhisa Sakai[11], Kazuhiro Yamakawa[12], Takuya Sasaki [4], Jun Nagai [3], Yoshiya Kawaguchi [13], Masaki Sone [10], Takayoshi Inoue [1], Yasuhiro Go [9,14,15], Noritaka Ichinohe[11], Kozo Kaibuchi[16], Masahiko Watanabe[17,18], Schuichi Koizumi [2], Michisuke Yuzaki [5] & Mikio Hoshino [1] ✉

In the central nervous system, astrocytes enable appropriate synapse function through glutamate clearance from the synaptic cleft; however, it remains unclear how astrocytic glutamate transporters function at peri-synaptic contact. Here, we report that Down syndrome cell adhesion molecule (DSCAM) in Purkinje cells controls synapse formation and function in the developing cerebellum. *Dscam*-mutant mice show defects in CF synapse translocation as is observed in loss of function mutations in the astrocytic glutamate transporter GLAST expressed in Bergmann glia. These mice show impaired glutamate clearance and the delocalization of GLAST away from the cleft of parallel fibre (PF) synapse. GLAST complexes with the extracellular domain of DSCAM. Riluzole, as an activator of GLAST-mediated uptake, rescues the proximal impairment in CF synapse formation in Purkinje cell-selective *Dscam*-deficient mice. DSCAM is required for motor learning, but not gross motor coordination. In conclusion, the intercellular association of synaptic and astrocyte proteins is important for synapse formation and function in neural transmission.

Neurotransmitters are released from the presynaptic terminal to the synaptic cleft and bind to their specific receptors on the postsynaptic terminal to transduce electrophysiological information. Free residual neurotransmitters are imported into the cytoplasm of the synapse or astrocytes enwrapping synapses through transporters embedded in the neuronal and astrocytic membrane[1]. The mechanism of transporter-mediated neurotransmitter clearance is tightly controlled because excessive or leaked neurotransmitters can activate either significant or inappropriate neural circuits, which can lead to mental illness and convulsion. However, how these functionally critical transporters localize to the synapses and function near the peri-synaptic area remains largely unknown.

The development of the mouse cerebellar nerve system is a good example of neuronal circuit formation, including synaptogenesis, dendritic formation, and pruning[2,3]. In Purkinje cells of the cerebellum, a single strengthened climbing fibre (CF) monopolizes proximal dendrites, whereas hundreds of thousands of parallel fibres (PFs) are projected from granule cells innervating distal dendrites. At the early

developmental stage (postnatal day 0 [P0] to P7) of mice, multiple CFs, which are the axons from the inferior olive, innervate a single Purkinje cell soma and make synaptic contact. Then, a single CF is strengthened depending on neural activity, translocates on Purkinje cell primary dendrite, and makes more synapses at a later stage from P9. Before CF translocation, Purkinje cell primary dendrites are connected to many PF terminals. Given that both dendritic translocations of the strengthened CF proceed upward from the base of the dendritic tree, the late phase of the CF synapse formation is critically dependent on the formation and/or elimination of PF synapses[4].

The molecules related to this CF elimination and translocation mechanism have been identified[4-11]. In addition to these molecules, the astrocytic glutamate-aspartate transporter (GLAST) is reportedly involved in CF synapse formation[12]. GLAST-knockout mice showed significantly weakened dendritic innervation by the main ascending CF[12]. GLAST plays an important role in the development and maintenance of proper synaptic wiring and wrapping in Purkinje cells[13,14]. Interestingly, GLAST is localised at the cell membrane of the Bergmann glia in the cerebellum but its distribution is not uniform but uneven. GLAST shows strong localisation to areas in contact with neuropils and synapses but localises weakly to other areas, such as the pia mater facing regions and dendrite shafts of the Purkinje cells[15]; however, the molecular mechanism underlying the uneven localisation of GLAST remains unclear.

Down syndrome cell adhesion molecule (DSCAM) is a cell adhesion molecule in the immunoglobulin superfamily that is essential for multiple aspects of neuronal wiring[16,17]. Various studies have revealed the importance of trans-homophilic binding between the same type of *Dscam* alternative splicing variants in flies[18,19] or the paralogs of *Dscam* in mice, in cellular functions such as axon guidance[19], dendrite arborisation[20], and synaptic contact[21-28]. Studies suggest that the heterophilic interaction of Netrin-1 and its receptor, deleted in colorectal cancer (DCC), with DSCAM mediates neurite outgrowth and commissural axon projection in neural development[29-31]. However, some genetic variants of *Dscam* existing in patients with autism spectrum disorder and schizophrenia[32-37] have been found in extracellular domains distinct from homophilic binding or Netrin-1/DCC binding; these studies implied the importance of heterophilic binding to other extracellular domains.

Here, we aimed to examine the functional role of DSCAM in synapse formation during cerebellar development by analysing the electrophysiological features and microstructure of synapses using electron microscopy. We examined the association of DSCAM and GLAST in the rough synaptosomal fraction and cell lysates overexpressing the extracellular domains of DSCAM. We also attempted to identify the DSCAM-expressing cells important for CF synaptogenesis by using four types of cerebellum-specific conditional knockout (cKO) mice. Using riluzole, which is reported to be an activator of GLAST-mediated uptake[38,39], we performed a rescue experiment of the defect in CF translocation and formation. Finally, we examined motor learning in the *Dscam*-mutant mice. These analyses are expected to provide valuable insights regarding the critical function of DSCAM through astrocytic GLAST in functional synapse formation in the developing cerebellum.

## Results

### DSCAM is required for CF synaptogenesis in the cerebellum

We first analysed the expression profiles of *Dscam* transcripts in each cell type of adult mouse cerebellum using public single-cell portal sites and high-throughput single-nucleus RNA-seq (snRNA-seq) data sets submitted by Kozareva et al[40]. As shown in Fig. 1a, prominent expression was observed in Purkinje cells and oligodendrocyte precursor cells (OPCs), but rarely in Bergmann glia and granule cells. These results were consistent with those of previous snRNA-seq analyses using the P8 mouse developmental cerebellum submitted by another

group[41] (Supplementary Fig. 1a) or the results of in situ hybridisation[42]. At P56, *Dscam* transcripts were also detected in inferior olive nucleus (ION) neurons projecting CFs (Fig. 1b), as well as in Purkinje cells and interneuron-like cells in the molecular layer. We then investigated the temporal expression profiles of DSCAM protein in the cerebellum during synaptogenesis using western blot analysis (Fig. 1c). Mouse cerebellum extracts were prepared from P0, the early stage of synaptogenesis in the cerebellum[3], to P28. DSCAM expression was observed at all stages examined, with a relatively higher expression at P0. This expression profile is similar to that of retinoic acid-related orphan receptor α (RORα), which is continuously expressed in Purkinje cells and molecular layer interneurons (MLI)[43], but its relative amount of RORα protein was decreased along with the proliferation of granule cells and a relative decline in cell number of Purkinje cells (Fig. 1c). Based on the results of cerebellar single-cell analyses and in situ hybridisation, DSCAM seems to be continuously expressed from the developmental stage to adulthood in Purkinje cells, ION neurons, and OPCs, but relatively less in the Bergmann glia and granule cells.

Next, we investigated DSCAM protein localisation using several anti-DSCAM antibodies and P15 and P30 mouse cerebellar slices via immunohistochemical analyses. Upon comparison of immunostained samples between control and *Dscam* conditional knockout (cKO) mice targeting all cell types of the caudo-dorsal midbrain and cerebellum (*En1^Cre^*; *Dscam ^flox/flox^*)[44,45], we observed high background staining in Purkinje cells, indicating poor antibody-specificity in the developing cerebellum, probably because of the presence of unknown antigens (Supplementary Fig. 1b–d). Thus, to investigate the detailed subcellular localisation of DSCAM, we electroporated pCAG–DSCAM–mEGFP together with hyperactive piggyBac transposase and transposon vectors (which express DsRed2 as a marker of transfection for labelling progenitor cells) to Purkinje cells in the cerebellum[46,47] (Fig. 1d, e). We previously reported that subcellular localisation, dimerising potential, or protein binding ability were not different between non-tagged DSCAM and DSCAM-mEGFP[48]. Following *in-utero* electroporation at embryonic day 12.5 (E12.5), we performed immunostaining of the cerebellum slice at P15 using an anti-EGFP antibody. DSCAM–mEGFP expression was low; however, an anti-GFP antibody enabled visualisation of its localisation (Fig. 1e). DSCAM–mEGFP was distributed throughout the dendrites and in the DsRed2-positive dendritic spine of Purkinje cells (Fig. 1e). DSCAM–mEGFP immunolabeling was also observed in a thin structure like the premature dendritic spine, which was relatively invisible with DsRed2 labelling. We also prepared the rough synaptosomal fraction using P20 mouse cerebella and confirmed that a strong accumulation of DSCAM was observed in the synaptosomal fraction similar to other synaptic proteins, GluD2 and PSD95 (Fig. 1f).

Next, we created a knock-in mouse line designed to express the DSCAM protein tagged with three consecutive ALFA[49] (Fig. 1g, h, Supplementary Fig. 2a–g). Weak but significant signals were detected in *Dscam^ALFA/ALFA^* mice, which were hardly observed in WT mice (Supplementary Fig. 2e). The DSCAM-ALFA signals were preferentially detected at the Calbindin-positive regions corresponding to the dendritic structures of Purkinje cells (indicated by yellow arrowheads in Fig. 1g). Additionally, a portion of DSCAM-ALFA signals was found in the vicinity of PSD95 (postsynapse marker; yellow arrowheads in Fig. 1h), vGluT1 (PF synapse marker; white arrows in Supplementary Fig. 2f), and vGluT2 (CF synapse marker; white arrows in Supplementary Fig. 2g). These immunoblot and immunohistochemical analyses suggest, at least in part, the localisation of endogenous DSCAM at synapses on Purkinje cells.

To understand the functional roles of DSCAM in the cerebellum, we examined the cerebellar constitution in mice carrying a spontaneous loss-of-function mutation in *Dscam* (*Dscam^del17/del17^*), with a 38-bp deletion in exon 17 of the *Dscam* gene[50]. The overall structure, cell placement, and cell density of P30 mutant cerebella were

 2

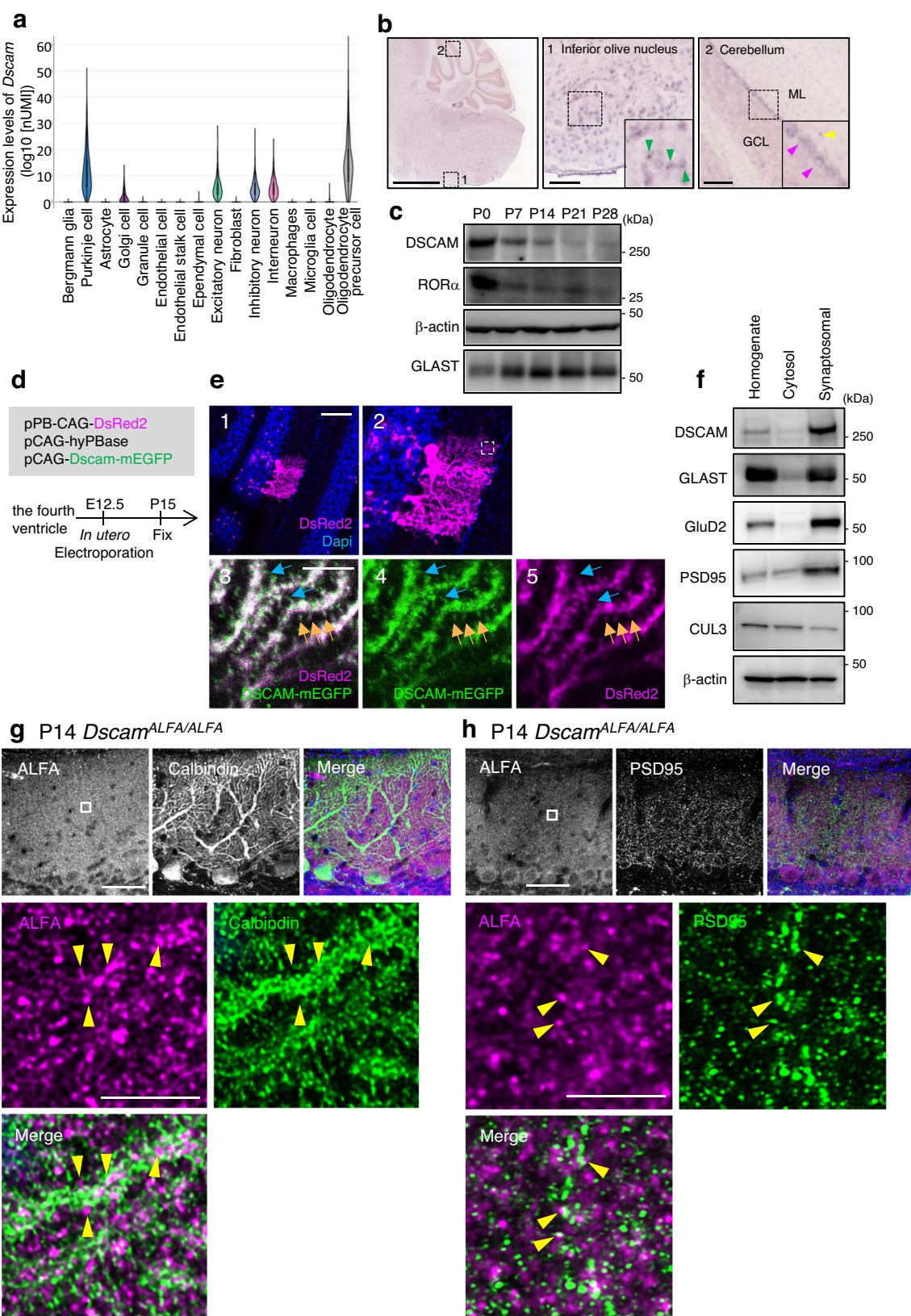

indistinguishable from those of wild-type (WT) mice, as reported previously[50] (Supplementary Fig. 3a, b). The distal portion of the caudal lobuli is unusually longer[50] compared to those of the WT from P7 to P30 (Supplementary Fig. 3b). The dorsal midbrain of $Dscam^{del17/del17}$ mice was reported to be hypertrophic[48,51]. Therefore, the enlarged midbrain may compress the cerebellum in the caudal direction,

resulting in stretching of the lobuli at the caudal side, although other possibilities remain. The height of the molecular layer was significantly lower than that of the littermate WT at P30 (Supplementary Fig. 3c).

Immunohistochemical analysis of P30 cerebellar slices with vesicular glutamate transporter 2 (vGluT2, a marker for presynaptic terminals of the CFs) showed a significant reduction in the vGluT2

**Fig. 1 | Expression and localisation of DSCAM in the developing mouse cerebellum. a** Violin plot of log10 (nUMI) per profile across the 16 cell types identified. The relative median values are consistent with known differences in cell size; e.g., Purkinje cells have the highest median number of UMIs. UMI: unique molecular identifier. **b** In situ hybridisation images of the mouse cerebellum and its surrounding area. Allen Brain Atlas, https://mouse.brain-map.org/gene/show/13287[93]. The green arrowhead indicates the signals in the inferior olive nucleus. The pink and yellow arrowheads indicate the signals of the Purkinje cells and interneurons, respectively. Scale bars in the left and middle/right boxes are 1.68 mm and 100 μm, respectively. **c** Temporal expression profiles of DSCAM protein in the cerebellum. Data are confirmed by three independent experiments. **d** Experimental design for *in-utero* electroporation and analyses. **e** P15 sagittal cerebellar sections expressing DSCAM−mEGFP with DsReds2 in electroporated Purkinje cells. **2**, High

magnification picture of **1**. Boxed regions indicate the area shown in **3–5**. **3–5**, Orange arrows indicate the dendritic spine of Purkinje cells. Blue arrows indicate DSCAM-EGFP-positive thin structures like filopodia. Scale bars **1** and **3** are 100 μm and 5 μm, respectively. Data are confirmed by three independent experiments. **f** Fractionation of DSCAM into a synaptosomal fraction using P20 wild-type cerebella. Data are confirmed by three independent experiments. Immunostainings of endogenous ALFA-tagged DSCAM and Calbindin (**g**) or PSD95 (**h**) in cerebellar slices from *Dscam^ALFA/ALFA^* mice on postnatal day 14 (P14). (top) Boxed regions are enlarged in the bottom images. Yellow arrowheads indicate adjacent localisation of DSCAM-ALFA and Calbindin (**g**) or PSD95 (**h**). Data are confirmed by three independent experiments. Scale bars, 50 μm (top), 5 μm (bottom). Source data are provided as the Source Data file.

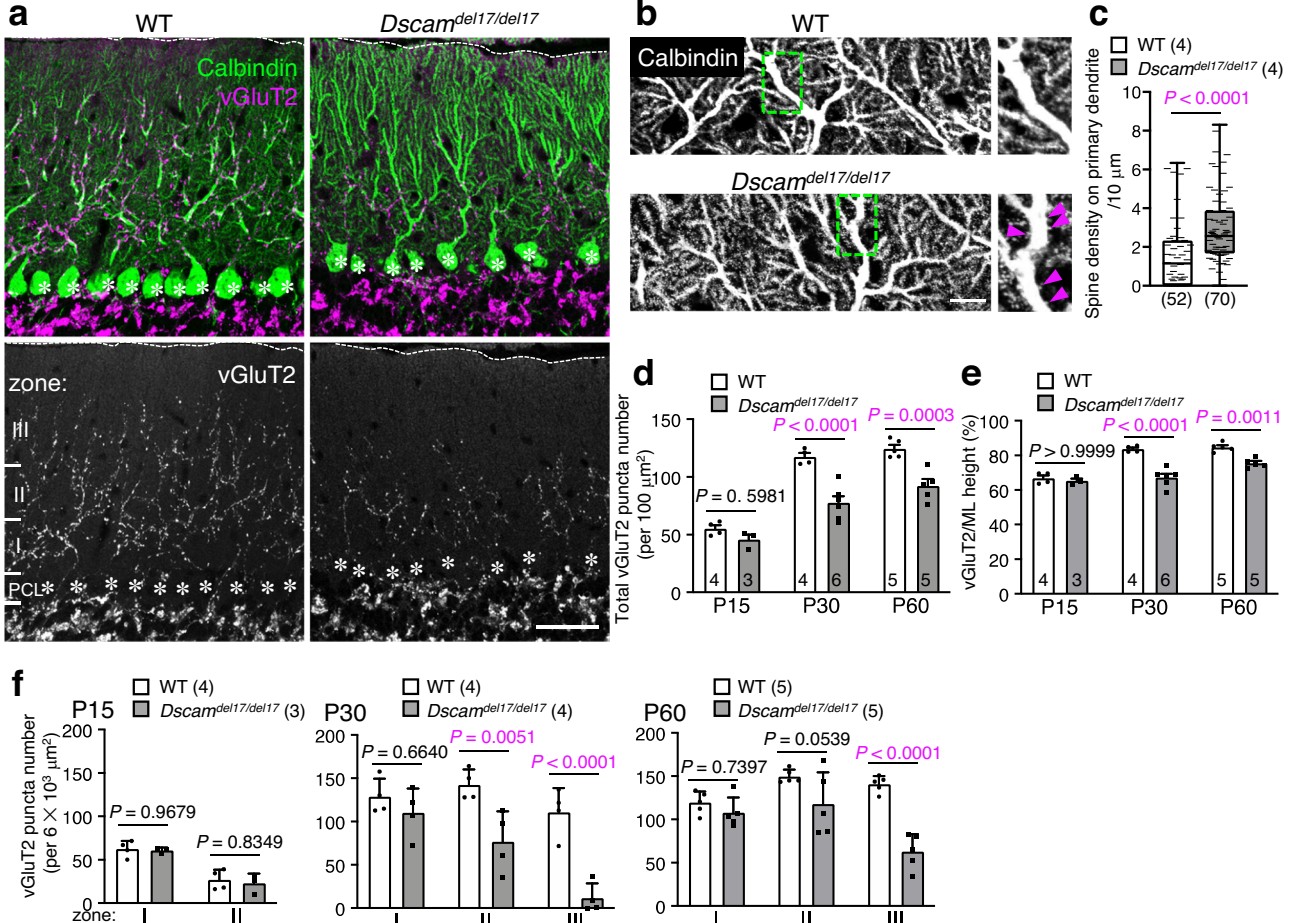

**Fig. 2 | Regressed CF territory and decreased CF synapses in *Dscam^del17/del17^* cerebella. a** Immunohistochemistry of Calbindin and vGluT2 in P30 wild-type (WT) (left) and *Dscam^del17/del17^* (right) mouse cerebellum. Dotted lines and asterisks represent the pial surface and Purkinje cell soma, respectively. The molecular layer located at the upper area of the Purkinje cell layer (PCL) was divided into three areas in **f**: Zone I was up to 40 μm in height from the PCL; Zone II was from 40 μm to 80 μm in height; Zone III was above 80 μm in height. Scale bar, 50 μm.
**b** Immunohistochemistry of Calbindin in P30 WT (upper) and *Dscam^del17/del17^* (lower) mouse cerebellum. The boxed regions are enlarged in the right images. Pink arrowheads represent ectopic spines from proximal dendrites. Scale bar, 50 μm.
**c** The density of spines on the primary dendrites per 10 μm dendrite length. WT,

*N* = 4 mice, *n* = 52 dendrites; *Dscam^del17/del17^*, *N* = 4 mice, *n* = 70 dendrites, box plots show median (horizontal line), quartiles (box), and range (whiskers), two-tailed Mann-Whitney test. **d** Developmental changes in the total vGluT2 puncta number per 100 μm². The numbers in each column indicate the number of mice examined. Data represent mean ± SEM; Two-way ANOVA with multiple comparisons.
**e** Developmental changes in the ratio of vGluT2 height per height of molecular layer (ML). The numbers in each column indicate the number of mice examined. Data represent mean ± SEM; Two-way ANOVA with multiple comparisons.
**f** Developmental and regional changes in the total number of vGluT2 puncta per 6 × 10³ μm². The zone was represented in **a**. Data represent mean ± SEM; Two-way ANOVA with multiple comparisons.

puncta in *Dscam^del17/del17^* cerebella compared to those in the WT slices (Fig. 2a, d, f). In P30 WT cerebella, CF terminals were translocated to dendrites and penetrated 80.2 ± 0.5% of the molecular layer thickness, whereas the depth reached by most distal CF terminals reduced significantly in *Dscam^del17/del17^* cerebella (62.7 ± 0.3%, Fig. 2a, e). Many

dendritic spines characteristic to PF terminals on primary dendrites of Purkinje cells, which normally disappear parallelly with CF synapse translocation from around P15 to P30[3], were still observed in the P30 *Dscam^del17/del17^* cerebella (Fig. 2b, c). The reductions in vGluT2-positive CF terminal number and height were not observed in *Dscam^del17/del17^*

cerebella at P15 and P22 but were evident at P30 and P60 (Fig. 2d, e, Supplementary Fig. 3d, e). The reduction in vGluT2-positive puncta was prominent in the middle and distal areas of the molecular layer at P30 (Fig. 2f). Although these vGluT2-puncta translocated upward in *Dscam*[del17/del17] cerebella until P60, they never reached the level observed in the control mice at P30 (Fig. 2f). These results suggested that DSCAM played a critical role in CF synaptogenesis and translocation after P22.

At several developmental stages (P7, P15, and P30), we performed immunostaining for various markers of specific cell types, synapses, or apoptosis (Supplementary Fig. 4, 5, Supplementary Table 1). During development (from P7 to P30), prominent changes were hardly observed in the morphology of Purkinje cells (Calbindin, a marker for Purkinje cells; Fig. 2a, Supplementary Fig. 4a, c, 5g), the density or distribution of vGluT1 (a marker for PF terminals; Supplementary Fig. 4a, b) or vesicular GABA transporter (vGAT, a marker for inhibitory fibre terminals; Supplementary Fig. 4c, d, 5g), or in the morphology of Bergmann glial cells (GFAP, a marker for Bergmann glia; Supplementary Fig. 4e, 5h). The number of Caspase-3-positive apoptotic cells (Supplementary Fig. 5a) and Ki67-positive proliferative cells (Supplementary Fig. 5b) remained unchanged in *Dscam*[del17/del17] cerebella at all the tested developmental stages. The number of Olig2-positive cells, including oligodendrocytes and OPCs, decreased significantly in *Dscam*[del17/del17] cerebella at P7 but not at P15 and P30 (Supplementary Fig. 5c, Supplementary Table 1). DSCAM may have some roles in the proliferation of OPCs, as suggested by the relatively high *Dscam* transcript levels in OPCs (Fig. 1a, Supplementary Fig. 1a).

## Dysfunction of GLAST in Dscam[del17/del17] cerebella

We evaluated CF synapse function in P30 *Dscam*[del17/del17] mice by assessing CF-evoked excitatory postsynaptic currents (CF-EPSCs) in Purkinje cells with whole-cell patch-clamp recordings (Fig. 3a–g). The number of functional CFs in contact with single Purkinje cells was estimated by varying stimulus intensities because a single CF input has a single threshold for excitation. Single CF-EPSCs were elicited in both WT and *Dscam*[del17/del17] cerebella at various stimulus intensities (Fig. 3a–c), suggesting that the Purkinje cells attained a one-to-one relationship with CFs in WT and *Dscam*[del17/del17] mice to a similar extent. The CF-EPSCs elicited by stimulation of *Dscam*[del17/del17] cerebella showed normal paired-pulse depression (Fig. 3d), indicating that the presynaptic functions were largely unaffected. Furthermore, there were no significant differences in basic electrophysiological parameters of CF-EPSCs, including the amplitude, rise time, EPSC area, and decay time constant, between WT and *Dscam*[del17/del17] mice (Fig. 3e–g, Supplementary Fig. 6a–f).

As for the PF-EPSCs, there was no change in any of the parameters examined between the WT and *Dscam*[del17/del17] mice (Fig. 3h–j, Supplementary Fig. 6g–l). A subtle delay in the recovery slope in the *Dscam*[del17/del17] cerebella was observed at low frequency. Thus, to estimate glutamate clearance by glutamate transporters, we used cyclothiazide (CTZ), which reduces the desensitisation of AMPA receptors, and therefore, unmasks the change in glutamate clearance of glutamate transporters[14,52]. In the presence of 100 µM CTZ, a significant difference was detected in the amplitude ratio of PF-EPSCs (Fig. 3h–j), but not in CF-EPSCs between the WT and *Dscam*[del17/del17] mice (Fig. 3e–g), suggesting partial hypofunction of glutamate clearance from the PF synapses in the *Dscam*[del17/del17] cerebella. The application of CTZ tended to increase the amplitude in the early phase, but not in the late phase, of PF-EPSCs in the *Dscam*[del17/del17] mice (Fig. 3h). Previous studies have shown that astrocytic glutamate transporter GLAST in the cerebellum contributes to the uptake of glutamate in the early stages after transmitter release[14]. In GLAST-deficient cerebella, CTZ treatment increased the amplitude of PF-EPSC at the early phase[14], and CF synapse translocation at the molecular layer was decreased[12], as observed in *Dscam*[del17/del17] mice (Fig. 2a). These electrophysiological

combined with previous GLAST KO data raised the possibility that glutamate uptake by GLAST in Bergmann glia is impaired at PF synapses in *Dscam*[del17/del17] mice.

To explore the anomalies of GLAST in *Dscam*[del17/del17] mice, we measured the quantities of *Glast* (*Slc1a3*) mRNA and GLAST protein in the cerebellar tissue of P21 WT and *Dscam*[del17/del17] mice by deep RNA sequencing (RNA-seq), western blotting, and immunohistochemical analyses (Supplementary Fig. 7a–d). There were no significant differences in the amount of GLAST transcripts and proteins between WT and *Dscam*[del17/del17] mice (Supplementary Fig. 7a). Immunohistochemical analyses revealed that in the *Dscam*[del17/del17] cerebellum, GLAST signals were localised along the Purkinje cell dendrite shaft similar to that of the WT (Supplementary Fig. 7d), and no significant difference was observed in the quantity of GLAST between WT and *Dscam*[del17/del17] mice (Supplementary Fig. 7c).

The cytoplasmic domain of DSCAM has been reported to exhibit transcriptional activity for genes involved in neuronal wiring in HEK293 cells[53]. Thus, we addressed the possibility that deregulation of the CF synapse is caused by the reduced expression of CF synaptogenesis-related genes, via RNA-seq analyses using the P21 whole cerebellum of WT and *Dscam*[del17/del17] mice (Supplementary Fig. 8a–j). Expression levels of most of the genes were not significantly different between the WT and *Dscam*[del17/del17] mice, probably because the population of *Dscam-expressing* cells in the cerebellum is quite low (compared to that of granule cells), including Purkinje cells (Fig. 1a). We also confirmed the unchanged expression of Purkinje cell-specific genes (and their encoded proteins), such as *Cacna1a* (Cav2.1)[10,11,54], *Sema7a* and *Sema3a*[5], *Grm1* (mGluR1) and *Prkcg* (protein kinase Cγ [PKCγ])[4], *Grid2* (GluD2)[55], and *Adgrb3* (Bai3)[9] between the WT and *Dscam*[del17/del17] mice (Supplementary Fig. 8a–j). These findings suggest the unlikely involvement of the transcriptional regulatory function of DSCAM in CF synapse development.

The membranous translocation of PKCγ and phosphorylation levels of myristoylated alanine-rich C-kinase substrate (MARCKS), which occurred upon mGluR1 activation mediating CF synapse elimination from dendritic spine[56,57], were not distinct between WT and *Dscam*[del17/del17] mice (Supplementary Fig. 9a–d).

If GLAST is disengaged from the peri-synaptic cleft, glutamate uptake from the synaptic cleft is impaired, which induces abnormal synaptic formation and functions[58]. Thus, we observed the ultrastructural morphology of the PF synapses and the GLAST localisation of P30 WT and *Dscam*[del17/del17] mice using electron and immunoelectron microscopy (Fig. 3k–m, Supplementary Fig. 10a–g). The densities of PF synapses were unchanged between the WT and *Dscam*[del17/del17] mice (Supplementary Fig. 10a, b). The size of the post-synapse did not change, but the pre-synapse size was significantly smaller in *Dscam*[del17/del17] mice than in WT mice (Supplementary Fig. 10c, d). There were no changes in the distance from the edges of postsynaptic density (PSD) to the closest Bergmann glia microfibres (Supplementary Fig. 10e, f), and the rate of glial attachment to the synaptic cleft (Supplementary Fig. 10g) between the WT and *Dscam*[del17/del17] cerebella. We then examined the distance from the GLAST molecules to the edge of the PSD using immunoelectron microscopic analyses (Fig. 3k–l); the distance in the mutants was significantly longer than that in WT mice. However, the density of immunogold particles was comparable between WT and *Dscam*[del17/del17] mice (Fig. 3m). These results suggested that the depletion of Dscam caused the delocalisation of GLAST away from the synaptic cleft.

## DSCAM binds to GLAST at the extracellular domain

Next, we investigated the mechanism by which DSCAM regulates GLAST localisation. Previously, we searched for novel DSCAM-binding molecules using whole-brain lysates by immunoprecipitation followed by mass spectrometry and identified candidate proteins[48]. Among them, the GLT-1a isoform (also known as EAAT2/Slc1a2), an astroglial

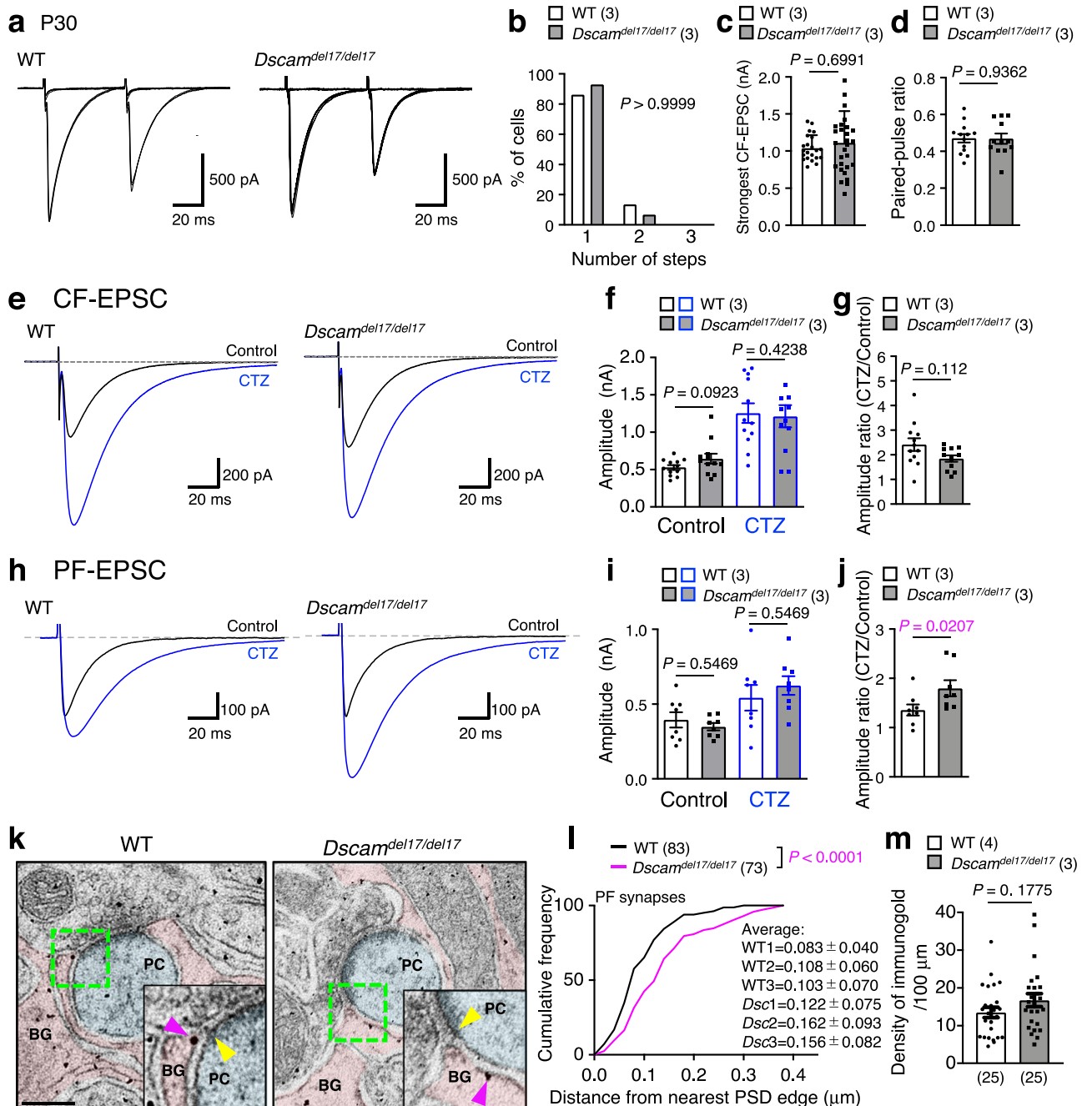

**Fig. 3 | Hypofunction of GLAST in PF synapse in *Dscam^{del17/del17}* mouse.**
**a** Representative traces of stimulated CF-EPSCs from P30 cerebellar slices in adult WT (left) and *Dscam^{del17/del17}* (right) mice. Superimposition of CF-EPSCs evoked by various stimulus intensities. CF: climbing fibre. **b** Frequency distributions of the number of CFs innervating each Purkinje cell at P30. The numbers in parentheses are mice count. Two-tailed Wilcoxon matched-pairs signed rank test. Strongest CF-EPSC amplitudes (**c**) and the paired-pulse ratio (**d**) from P28–32. The numbers in parentheses are mice count. Statistical analysis was performed using data from each neuron indicated by the dots in the graph. Data represent mean ± SEM with individual data from each neuron (**c** WT, $n = 22$; *Dscam^{del17/del17}*, $n = 29$, **d** WT, $n = 13$; *Dscam^{del17/del17}*, $n = 12$); two-tailed Mann-Whitney $U$ test. Datasets of CF-EPSCs (**e**–**g**) and PF-EPSCs (**h**–**j**). Representative traces of EPSCs (**e** and **h**), amplitude (**f** and **i**), and amplitude ratio (**g** and **j**) from P28–32. The numbers in parentheses are mice count. Statistical analysis was performed using data from each neuron indicated by the dots in the graph. Data represent mean ± SEM with individual data from each neuron (**f**–**g**, WT, $n = 12$; *Dscam^{del17/del17}*, $n = 12$, **i**–**j**, WT, $n = 8$; *Dscam^{del17/del17}*, $n = 8$);

two-tailed Mann-Whitney $U$ test (**f**–**g**, **i**–**j**). PF-EPSC: parallel fibre-evoked excitatory postsynaptic current. CTZ: cyclothiazide. CF-EPSC: climbing fibre-evoked excitatory postsynaptic current. **k**, Immunoelectron microscopy of WT and *Dscam^{del17/del17}* mouse molecular layer. Bergmann glial (BG) processes and dendritic spines of Purkinje cells (PC) are tinted pink and light blue, respectively. Higher magnification in the bottom-right corner represents the area surrounded by the green dotted box. The nearest postsynaptic density (PSD) edge and closest GLAST labelled with immunogold are indicated by yellow and pink arrowheads, respectively. Scale bar, 200 nm. **l**, Cumulative histogram of the distance between nearest PSD edge to closest GLAST labelled by immunogold in PF synapse. Samples were collected from three littermates per genotype. WT, $n = 86$ PSD edges; *Dscam^{del17/del17}*, $n = 75$ PSD edges. Unpaired two-tailed t-test. **m**, The density of GLAST labelled by immunogold. WT, $N = 4$ mice, $n = 25$ Bergmann glia; *Dscam^{del17/del17}*, $N = 3$ mice, $n = 25$ Bergmann glia. Statistical analysis was performed using data from each neuron indicated by the dots in the graph. Data represent mean ± SEM; two-tailed Mann-Whitney $U$ test.

glutamate transporter highly expressed in the forebrain, but relatively lower in the cerebellum[59], was identified as a DSCAM-interacting protein[48]. GLT-1 is highly homologous to GLAST in primary amino acid sequences[60]. Thus, we performed an immunoprecipitation assay using anti-GLAST and anti-DSCAM with the rough synaptosomal fraction of mouse cerebellum (Fig. 1f), addressing the in vivo association of DSCAM and GLAST (Fig. 4a, Supplementary Fig. 11a, b). We confirmed that GLAST and DSCAM co-immunoprecipitated reciprocally (Fig. 4a, Supplementary Fig. 11a, b). In $Dscam^{ALFA/ALFA}$ mice, GLAST was found to be co-immunoprecipitated selectively with anti-ALFA nanobody; however, this was not observed in WT mice lacking DSCAM-ALFA (Fig. 4b). To examine the intercellular association, we investigated the binding of GLAST and the extracellular domain of DSCAM by a co-immunoprecipitation assay using deletion constructs and COS7 cells (Fig. 4c,d). Deletion constructs lacking the Ig1-9 domain and the intercellular domain (ΔCΔIg) fused with mEGFP (ΔCΔIg-mEGFP), and full-length DSCAM (Full-mEGFP) and deletion constructs lacking the intercellular domain (ΔC-mEGFP) were associated with GLAST-flag (Fig. 4c, d), indicating that the extracellular region of DSCAM binds to GLAST. Some of the immunolabellings of endogenous DSCAM-ALFA and GLAST were localised adjacent to each other at the molecular layer in the cerebellum (Fig. 4e). DSCAM-ALFA-labelled structures seem to connect with the GLAST-labelled microfibres of Bergmann glia.

### Purkinje cell-specific Dscam cKO mice showed a reduced number of CF synapses

$Dscam$ was expressed in Purkinje cells, some interneurons in the cerebellum, and ION neurons (i.e., CF neurons) in the medulla oblongata (Fig. 1a, b). Since gene expression was deleted in all these cells in $Dscam^{del17/del17}$ mice, it is unclear which cells expressing DSCAM are responsible for CF synapse formation and translocation. Therefore, we created hindbrain-cell type-specific $Dscam$ cKO mice by crossing $Dscam$-floxed ($Dscam^{flox}$) mice[61] with various types of $Cre$ lines: $En1$-Cre, targeting all cell types of the caudo-dorsal midbrain and cerebellum from E8 ($En1^{Cre}$-cKO)[44,45]; $L7/Pcp2$-Cre, targeting Purkinje cells from P4 ($Pcp2^{Cre}$-cKO)[62]; $Ptf1a$-Cre, targeting Purkinje cells, interneurons, and ION neurons from E11 ($Ptf1a^{Cre}$-cKO)[63]; and $Tg$-Atoh1-Cre, targeting granule cells from E11 ($Atoh1$-Cre-cKO)[64] (Fig. 5a–e). $En1^{Cre}$-cKO mice showed smaller body size and ataxia to a lesser extent than $Dscam^{del17/del17}$ mice[51]. The other three cKO mouse strains showed no ataxia or nystagmus. The whole cerebellar structure, lobe architecture, and molecular layer height in all four of these mice strains were normal (Supplementary Fig. 12a–c). Immunohistochemical analysis of P30 cerebellar slices revealed that the number of vGluT2-positive puncta was significantly reduced in three mouse strains: $En1^{Cre}$-cKO, $Pcp2^{Cre}$-cKO, and $Ptf1a^{Cre}$-cKO compared to that in the control $Dscam^{flox/flox}$ mice (Fig. 5b, c). The translocation rate in the molecular layer (vGluT2 height/molecular layer height) was significantly decreased in $En1^{Cre}$-cKO mice, but not in other $Dscam$ cKO mice (Fig. 5d). Purkinje cell-specific depletion of $Dscam$ ($Pcp2^{Cre}$-cKO) reduced the number of vGluT2-positive CF synapses from proximal to distal areas, to a similar degree as that of $Ptf1a^{Cre}$-cKO, in which $Dscam$ was deleted in Purkinje cells and most cell types, except for the granule cells[63] (Fig. 5e). These results suggest that DSCAM expressed in Purkinje cells is, at least partially, critical for CF synapse formation during postnatal cerebellum development.

### Drug-induced glutamate reduction from the synaptic cleft rescued the CF synapse formation

To explore the impairment of glutamate transporter-mediated glutamate clearance in $Dscam$ mutant mice during CF synaptogenesis, we used riluzole, a popular medication used for amyotrophic lateral sclerosis (ALS)[65,66]. Riluzole acts as a neuroprotective drug to block the pathological influx of sodium in spinal cord injury[67] as well as an anti-glutamatergic agent[68] via the reduction of glutamate release[69].

Notably, Fumagalli et al. reported that riluzole significantly increased glutamate transporter-mediated uptake in a dose-dependent manner[38,39]. Thus, we used riluzole (13 mg/kg)[70,71] to activate the residual GLAST and to reduce the concentration of free glutamate at the synaptic cleft in Purkinje cell-specific Dscam cKO mice ($Pcp2^{Cre}$-cKO mice) (Fig. 6a–c). Intraperitoneal administration of riluzole was started from P15 (Fig. 6a), and we performed immunostaining of the cerebellar slices at P30 using an anti-vGluT2 antibody (Fig. 6b, c). Riluzole administration in $Pcp2^{Cre}$-cKO mice partly but significantly rescued CF synaptogenesis at the proximal area of the molecular layers (Fig. 6b, c). These results suggest that the excessive free glutamate at the synaptic cleft, probably caused by the delocalisation of GLAST apart from the peri-synaptic area, accounts for the impairment of CF synaptogenesis during postnatal cerebellum development.

### Functional significance of DSCAM in motor learning

To further clarify the role of DSCAM in Purkinje cells, we examined the adaptation of the horizontal optokinetic response (hOKR), which has been utilised as an experimental model for studying cerebellum-dependent motor learning[72] (Fig. 7a). Continuous oscillation of a screen around a stationary animal increased the hOKR gain in the control $Dscam^{flox/flox}$ mice (Fig. 7b–d). Although the hOKR was similarly observed in $Pcp2^{Cre}$-cKO mice before the training, the gain did not increase in $Pcp2^{Cre}$-cKO mice during the 60 min hOKR training (Fig. 7b–d), indicating that motor learning was severely impaired in Purkinje cell-specific $Dscam$-deficient mice. These findings indicate that the DSCAM protein expressed in Purkinje cells is required for normal CF functions, such as motor learning, in the adult cerebellum.

## Discussion

DSCAM is essential for synaptogenesis not only in the $Drosophila$ nervous system but also in the vertebrate brain[28,73]. The trans-synaptic homophilic interaction of the DSCAM family of proteins or their splice variants between presynapse and postsynapse is also important for the synaptic initiation and stabilisation[20,22,23]. In this study, we clearly demonstrated the significance of DSCAM function in CF synapse formation required for motor learning using a variety of cerebellar-specific conditional knockout mice (Supplementary Fig. 13a–c). We found that the intercellular heterophilic association of DSCAM with GLAST between postsynapse and enwrapped astrocytic Bergmann glia was required for proper CF synapse formation. However, there remains the possibility that trans-synaptic homodimer of DSCAM may also be formed in CF synapse because presynaptic ION neurons expressed $Dscam$ transcripts to some extent (Fig. 1b). The defects of CF synapse formation were comparable between postsynaptic Purkinje cell-specific cKO ($Pcp2^{Cre}$-cKO) mice and presynaptic ION neuron and postsynaptic Purkinje cell cKO ($Ptf1a^{Cre}$-cKO) mice (Fig. 5a–e). Thus, there is a possibility that trans-synaptic homodimer DSCAM in CF synapse may play a role in CF synapse translocation. We did not examine this possibility for lack of a proper tool to express DSCAM protein selectively to postsynaptic dendritic spines of CF synapse. However, we believe that the GLAST function in PF synapse is, at least in part, required for CF synapse formation, because the application of the anti-glutamatergic agent riluzole rescued the abnormal CF synaptogenesis induced by Purkinje cell-specific depletion of $Dscam$ in vivo (Fig. 6a–c). In addition, it has been repeatedly reported that abnormal PF synapse input and/or formation causes impairment of CF synapse translocation and formation (see below)[3,4,74,75]. Further analyses are required to elucidate the exact role of the trans-synaptic homodimer DSCAM in CF synapses.

### Mechanisms underlying DSCAM-mediated remodelling of CF synaptogenesis

The impairment of CF synapse translocation in the $Dscam^{del17/del17}$ cerebellum was prominent from P15. PF synapse activity is required

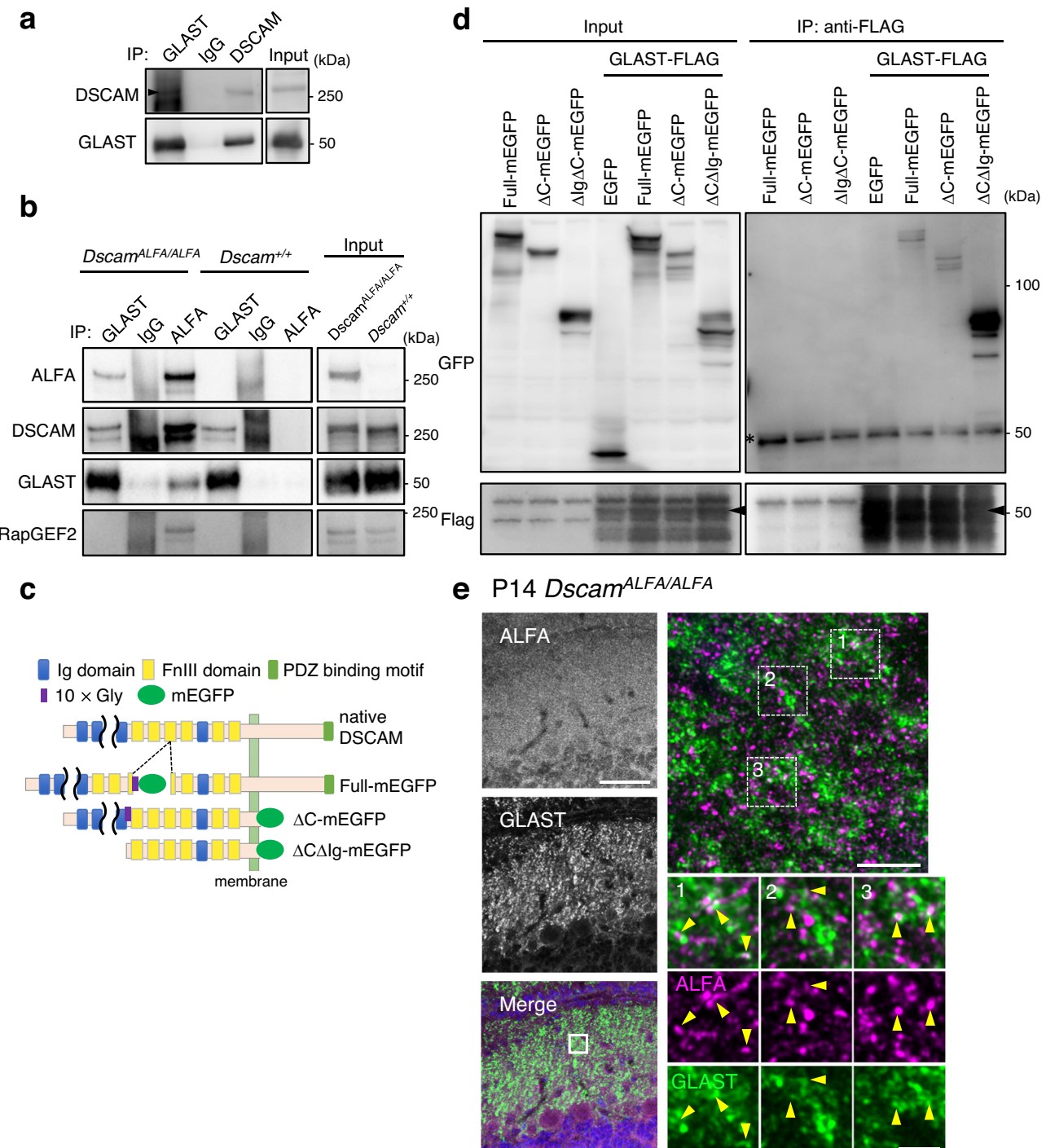

**Fig. 4 | DSCAM associates with GLAST. a** Co-immunoprecipitation and western blot assays using synaptosomal fraction (*n* = 3). The arrowhead indicates the bands of DSCAM. **b** Co-immunoprecipitation assays using whole brain lysate from *Dscam^ALFA/ALFA^* and *Dscam^+/+^* mice. The lysates were incubated with anti-GLAST or anti-ALFA nanobody and the immunoprecipitants analysed by western blot using anti-ALFA nanobody, anti-DSCAM, and anti-GLAST and anti-RapGEF2 antibodies. Data are confirmed by three independent experiments. Input, 1% of total lysate. **c** Diagram of the domain structure of native DSCAM and deletion constructs fused with mEGFP. **d** Co-immunoprecipitation assay using lysates of COS-7 cells expressing Dscam constructs and full-length GLAST fused with FLAG at the C-terminus were incubated with an anti-Flag antibody. The input (left: 5%) and

immunoprecipitants (IP, right) were analysed by western blotting using anti-GFP and anti-Flag antibodies. The experiment was repeated at least three times. The asterisk indicates the bands of antibodies. The arrowheads indicate the intact band of GLAST-FLAG. **e** Immunostainings of endogenous ALFA-tagged DSCAM and GLAST in cerebellar slices from *Dscam^ALFA/ALFA^* mice on postnatal day 14 (P14). The left images show wide-field views of ML. Scale bar, 50 μm. The boxed region in the merged image is enlarged in the top right corner image. Scale bar, 5 μm. High-magnification images for three rectangles (1, 2, 3) in the right upper panel are shown below. Yellow arrowheads indicate adjacent localisation of DSCAM-ALFA and GLAST. Data are confirmed by three independent experiments. Scale bar, 1 μm. Source data are provided as the Source Data file.

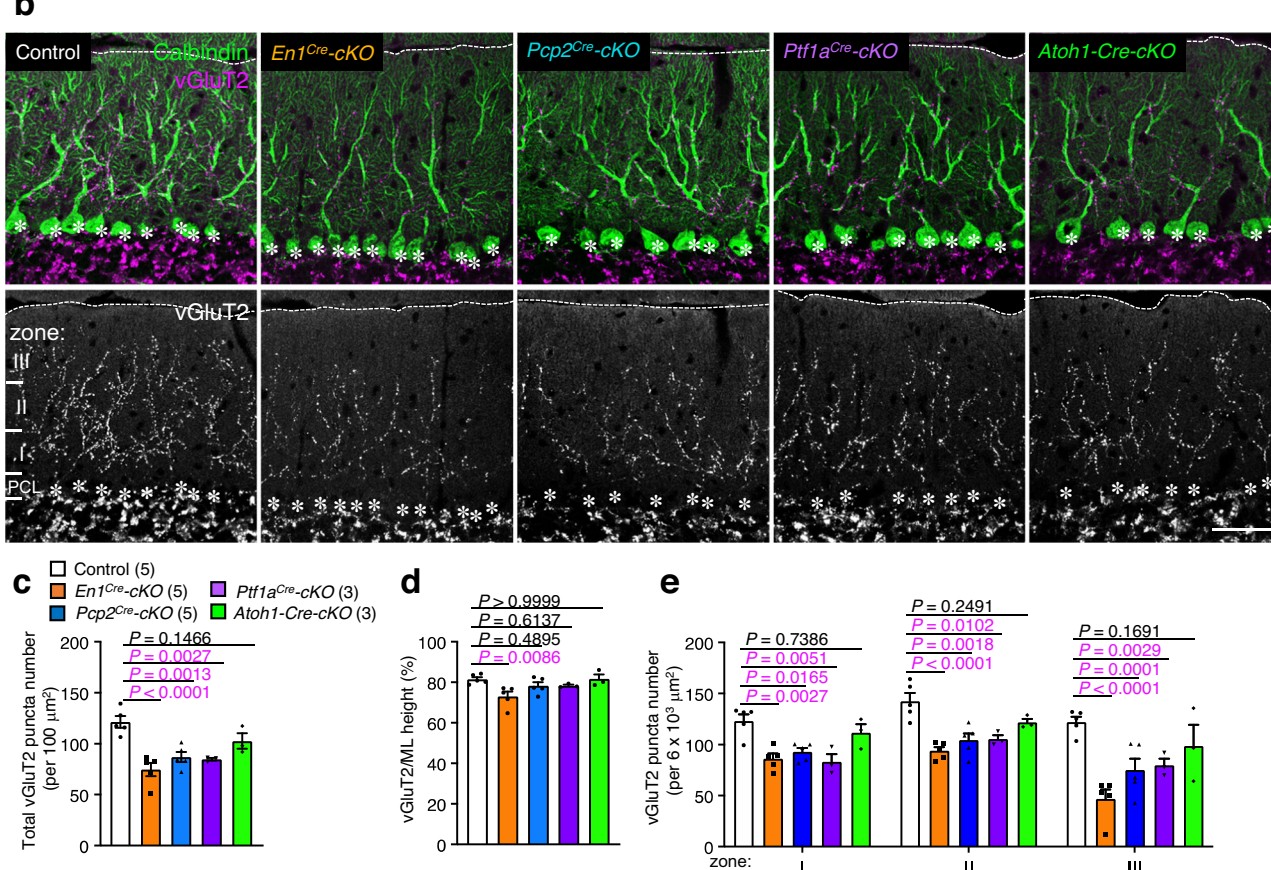

**Fig. 5 | CF synapse translocation is perturbed in conditional KO mice lacking _Dscam_ in Purkinje cells. a** Table showing the depletion of _Dscam_ on Purkinje cells, granule cells, and ION neurons in each cKO mouse. ION: inferior olive nucleus. CF: climbing fibre. cKO: conditional knockout. **b** Immunohistochemistry of Calbindin and vGluT2 in P30 Control and conditional KO mouse cerebella. Dotted lines and asterisks represent the pial surface and Purkinje cell soma, respectively. The molecular layer located at the upper area of the Purkinje cell layer (PCL) was divided into three areas in **e**; Zone I was up to 40 μm in height from the PCL; Zone II was

from 40 μm to 80 μm in height; Zone III was above 80 μm in height. Scale bar, 50 μm. Quantification of the total vGluT2 puncta number (**c**) and the ratio of vGluT2 height to molecular layer height (**d**) in the molecular layer of each cKO cerebella. The numbers in parentheses are mice count. Data represent mean ± SEM; Two-way ANOVA with multiple comparisons. **e** Regional changes in the total number of vGluT2 puncta per $6 \times 10^3$ μm². The zone was represented in **a**. The numbers in parentheses are mice count. Data represent mean ± SEM; Two-way ANOVA with multiple comparisons.

for heterosynaptic competition between PF and CF synapses from P15 onwards[4]. The territory overlaps and the subsequent segregation of PF and CF synapses on Purkinje cell proximal dendrites are driven by the elimination of PF synapses from the overlapping portions as well as dendritic translocation of the single strengthening CF from P15 to P30. In _Dscam_[del17/del17] mice, CF translocation weakened significantly along with the unusual formation of PF synapse-like cobbled dendritic spines at the proximal dendrites of Purkinje cells (Fig. 2a–c). Territory transition is arrested in mutant mice lacking mGluR1–Gαq–PLC4β–PKCγ signal transduction, which is important for input from PF synapses[4]. The GluD2–CBLN1–NEUREXIN system is also critical for the formation and maintenance of PF synapses, and thus, contributes to the establishment of properly segregated CF

territories[74,75]. Although we could not detect the unusual change in mGluR1 for PKCγ signalling or the changes in CF translocation-related transcript expression in Purkinje cells in _Dscam_[del17/del17] cerebella (Supplementary Fig. 8a–j, 9a–d), these reports implied that abnormal PF synapse input and/or formation caused impairment of CF synapse translocation and formation in _Dscam_[del17/del17] mice. Therefore, one possible hypothesis is as follows: impaired binding of DSCAM-GLAST at PF synapses promotes an increase in free glutamate at synaptic cleft and gives excessive input to the post-synapse of Purkinje cells. This excessive input may somehow inhibit pruning on primary dendrites at PF synapses and cause abnormal stabilisation of PF synapses, thereby preventing dendrite CF translocation and enhancing CF formation from P15 to P30.

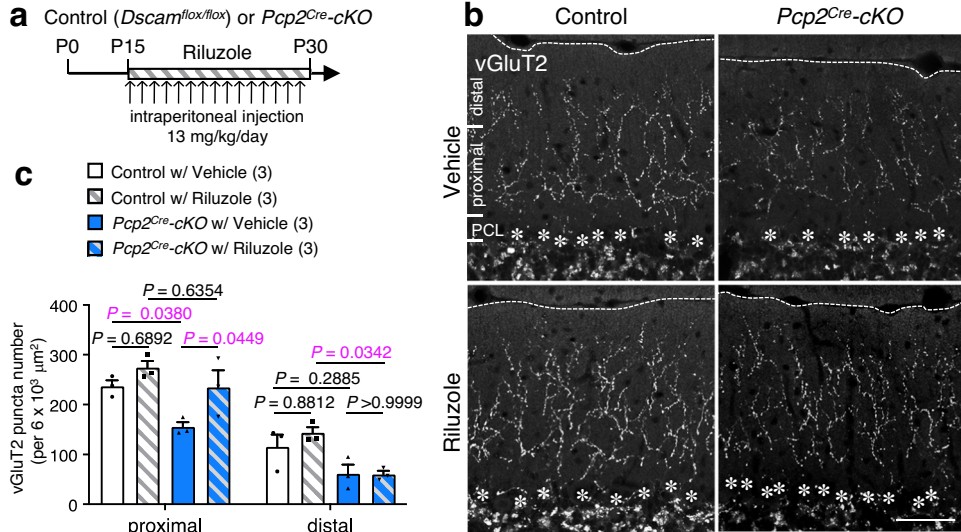

**Fig. 6 | Riluzole administration rescues the impaired CF translocation.**
**a** Experimental design for riluzole administration. CF: climbing fibre. Control: *Dscam^flox/flox^*. cKO: conditional knockout. **b** Immunohistochemistry of vGluT2 in adult control (left) and *Pcp2^Cre^-cKO* (right) mouse cerebella treated with vehicle (upper) or riluzole (lower). Dotted lines and asterisks represent the pial surface and Purkinje cell soma, respectively. The molecular layer located at the upper area of

the Purkinje cell layer (PCL) was divided into two areas in **c**. The proximal area was up to 80 μm in height from the PCL; the distal area was above 80 μm in height. Scale bar, 50 μm. **c** Quantification of vGluT2 puncta number in the proximal and distal half areas. The zone was represented in **b**. The numbers in parentheses are mice count. Data represent mean ± SEM; Two-way ANOVA with multiple comparisons.

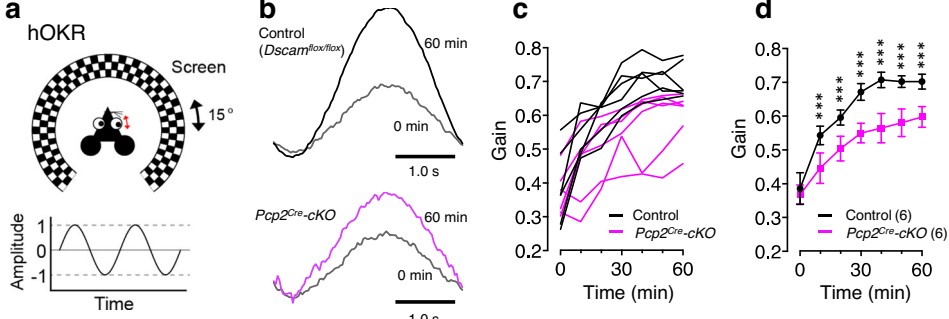

**Fig. 7 | Motor learning is impaired in conditional KO mice lacking *Dscam* in Purkinje cells.** **a** Cartoon of hOKR system. The mouse was placed on a table surrounded by a checkered screen, and the head was fixed. Eye movements were monitored while the screen oscillated sinusoidally. **b** Representative hOKR waveforms in *Pcp2^Cre^-cKO* (lower) and its control littermate (upper) mice before (0 min)

and after (60 min) training. Individual data (**c**) and the averaged values (**d**) of hOKR acquisition for 60 min in *Pcp2^Cre^-cKO* mice versus their control littermates. The numbers in parentheses are mice count. Data represent mean ± SEM; ***$p < 0.001$, Two-way repeated measures ANOVA in (**d**). Control: *Dscam^flox/flox^*.

## Differential effects on PF or CF synapses due to the depletion of *Dscam*

In this study, the application of CTZ significantly increased the amplitude ratio of PF-EPSCs in *Dscam^del17/del17^* mice, although the amplitude was not significantly affected. In this experimental system, there was a large variation in amplitude among different stimulus locations. That might be why we could not detect a significant difference in amplitude. However, we successfully observed a significant difference by taking amplitude ratios of PF-EPSCs before and after CTZ administration, which removed the contribution of AMPAR desensitization in each response and allowed a more direct assessment of the contribution of Glu clearance (Fig. 3e–j). In GLAST-KO mice, CTZ treatment increased the peak amplitude of the PF- and CF-EPSCs and prolonged its decay beyond the value observed in WT mice[14,76], which was not observed in *Dscam^del17/del17^* mice. The reason for this phenotypic discrepancy between *Dscam^del17/del17^* and GLAST-KO mice is probably that the control of GLAST by DSCAM is restricted to the peri-synaptic area. Usually, GLAST is localised not only in the inter-synaptic cleft but

also in the gap between the pre- or post-synapse and Bergmann glia, where it is thought to prevent spillover of superfluous glutamate to neighbouring synapses[76]. Impaired glutamate uptake has been implicated in neurotoxicity and neurological diseases[77,78]. In *Dscam^del17/del17^* mice, however, we observed no significant cerebellar atrophy due to cell death and no significant change in the number of Caspase3-positive cells, a marker of apoptosis (Supplementary Fig. 5a, Supplementary Table 1), and the amount of membranous GLAST was comparable to that of WT (Fig. 3m, Supplementary Fig. 7a–d). Therefore, the complete loss of GLAST may markedly increase superfluous free glutamate and EPSC amplitudes, whereas partial reduction of GLAST near the peri-synaptic area by loss of DSCAM may not induce a dramatic effect on EPSCs. The decrease in CF synapse seen in both GLAST KO and *Dscam^del17/del17^* mice may be due to the heterosynaptic competition between PF and CF synapses, as described above, and not due to toxicity from excess free glutamate.

We also found that the depletion of *Dscam* had a selective effect on PF-EPSCs, but not on CF-EPSCs (Fig. 3e–j). However, the application

of CTZ reportedly affects both PF and CF-EPSCs in GLAST-KO mice[14,76]. A possible explanation is that DSCAM is not a major regulator of GLAST in CF synapse stability. As we mentioned above, the ION neurons bearing CF also expressed *Dscam* (Fig. 1b), implying trans-synaptic DSCAM homodimer formation between the dendritic spines of Purkinje cells and axon terminals of ION neurons at the CF synapse[22,30,50]. Therefore, the DSCAM–GLAST interaction may not be functional in the CF synapses.

In *Pcp2^Cre^-cKO*, we found the defect in motor learning (Fig. 7a–d) but not in motor coordination such as during walking, grasping food, and running in daily life. Similar results have been reported previously. mGluR1b-rescue mice, wherein the splice variant of mGluR1 is expressed in mGluR1-null, exhibited impairments in CF synapse elimination, induction of long-term depression (LTD), and delay in eyeblink conditioning, another form of cerebellum-dependent motor learning but showed normal motor coordination[79]. Similarly, a single recombinant CBLN1 injection transiently restored PF- LTD and eyeblink conditioning in adult *Cbln1*-null mice without affecting the CF innervation pattern[80]. In *C1ql1*-null and PC-*Bai3*-null mice, motor learning and PF-LTD were severely impaired but motor coordination was normal[9]. Taken together, PF-LTD and motor learning, but not motor coordination, seem to be closely correlated[81]. Therefore, future studies should assess whether and how DSCAM proteins mediate synaptic plasticity in PF, CF, and other synapses essential for memory and learning.

## Astrocyte-synapse interactions play a definitive role in synapse development

Despite the importance of astrocyte-selective recognition of neuronal synapses in tripartite synapses, data concerning contact-mediated signalling between astrocytic and synaptic proteins are limited[82,83]. Interestingly, most identified contact-mediated astrocyte-neuron communication mediates the homophilic association of cell adhesion molecules, such as gamma protocadherins[84] and NRCAM[85], and promotes the formation of both excitatory and inhibitory synapses. EphrinA3/Eph4A[86,87] and Neuroligin/Neurexin[88] are also known as mediators of astrocyte-neuron interaction for synapse formation and function. Here we identified the intracellular heterophilic association between neuronal adhesion molecules DSCAM and astrocytic GLAST in the cerebellum, and the regulatory machinery of the peri-synaptic localisation of GLAST. This DSCAM-GLAST interaction in PF synapse was not only required for PF synapse formation and function (Figs. 2b, c, 3j, Supplementary Fig. 10a–d), but was also critical for the CF synapse formation (Figs. 2a, 5a–e, 6b, c). Given the diversity of the transmembrane proteins in the tripartite synapses, our present study sheds light on the intriguing possibility that astrocyte-neuron interaction provides not only mutual reciprocity for local synaptogenesis but also selective signals influencing the whole cell morphogenesis and wiring during the development and adulthood.

Astrocytes also contribute to memory, neuroprotection, and homeostasis via energy metabolism in the brain[89]. Glutamate taken up by astrocytes is used in the TCA circuit[90], and some of it is converted to α-ketoglutarate by glutamate dehydrogenase and utilised for ATP production in the mitochondria. This suggests that the delocalisation of GLAST may cause an energy deficit associated with impaired glutamate retrieval in the cerebellum. Therefore, further studies may be required to investigate phenotypes caused by abnormal energy metabolism in the *Dscam* cKO cerebella.

## DSCAM-GLAST interaction and neuropsychiatric disorders

To date, the DSCAM molecule has been known implicated in neuropsychiatric disorders such as schizophrenia and autism[32–37]. Many molecules involved in such disorders have been reported to be involved in synaptogenesis and transmission[33–37]. Among them, impaired glutamate removal is known to cause abnormal synapse formation, neuronal cell death due to excessive input, and behavioural abnormalities in mice[91]. Genetic mutations in glutamate transporters have also been reported in human genome wide sequencing of psychiatric disorders[33,34]. These results suggest that the interaction between DSCAM and GLAST is involved in neuropsychiatric disorders. Whether the localisation of GLT-1, another glutamate transporter, is also regulated by DSCAM is a subject for future study. In the future, we should focus on how these DSCAM functions are regulated throughout the brain regions.

# Methods

## Animals

All animal experiments in this study were approved by the Animal Care and Use Committee of the National Institute of Neuroscience, Japan (#2019027). Pregnant ICR mice were used for *in-utero* electroporation. *Dscam^del17^* mice[50] obtained from the Jackson Laboratory (JAX stock #008000) were used in our laboratory by crossing BALB/cCrSlc (BALBc) mice to improve the birth rate of *Dscam^del17/ del17^* mice. The *Dscam^flox^* allele has been described previously[61]. *En1-Cre*[44] mice were obtained from Dr. Alexandra L. Joyner (New York University School of Medicine). The *En1-Cre*; *Dscam^flox/flox^* (*En1^Cre^-cKO*) mice died neonatally in the C57BL/6 background; the experiments described were conducted on a mixed but predominantly BALB/cByJ background. The *Pcp2-Cre*[62] allele was obtained from the Jackson Laboratory (JAX stock #004146). The *Ptf1a-Cre*[63] and *Tg-Atoh1-Cre* alleles have been described previously[92]. The littermates were used as controls, and both sexes were used for the control and mutant mice in this study. Two or three pregnant mice were housed per cage and maintained on a 12-h light/dark cycle with free access to food and water. At the end of the experiment, the animals were deeply anesthetized with 2% inhaled isoflurane and then euthanized by perfusion fixation or cervical dislocation.

## DNA constructs

Complementary DNAs (cDNAs) encoding the full-length mouse *Dscam*, *Dscam–mEGFP* subcloned into a pCAG vector, were generated as described previously[48]. The procedure to generate DSCAM-mEGFP expression plasmid has been described previously[48].

Dscam-ΔC (1–1,615 aa) was amplified via PCR using PrimeSTAR Max (R045A; Takara Bio, Dalian, China) and the following primers: 5′- TTTTGGCAAAGAATTCATGTGGATACTGGCTCTCTCCTTGTTCCAGAGCTTCGCGAATGTTTTCAGTGAAGAGCCCCACTCCCCCCCCAGATCCTCCCGAG-3′ and 5′- GTCGACTGCAGAATTCGCAACCAGCAGAAGCACAAAGA-3′ (the sequence of the *N*-terminal signal peptide sequence is underlined). *Dscam–ΔCΔIg* (885–1,615 aa) was amplified via PCR using PrimeSTAR Max and the following primers, 5′- TTTTGGCAAAGAATTCATGTGGATACTGGCTCTCT-3′ and 5′- GTCGACTGCAGAATTCGCAACCAGCAGAAGCACAAAGA-3′. The obtained products were subcloned into the pCAG–mEGFP vector using the In-Fusion HD Cloning Kit (Clontech Laboratories, Mountain View, CA, USA). All sequences used for the construction of the DNA plasmid were sequence-checked. Hyperactive piggyBac transposase (pCAG–hyPBase) and piggyBac transposon vectors (pPB-CAG-DsRed2)[46,47] were kindly provided by Dr. Ryohei Yasuda (Max Planck Florida Institute for Neuroscience). pcDNA3.1-Slc1a3-DYKDDYK (mouse *Glast* full-length tagged with FLAG at the C-terminal, CMV promotor dependent) was obtained from GenScript (#OMu17704; Piscataway, NJ, USA).

## Rigorous selection/optimisation of the tag-insertion site in DSCAM

The selection and optimisation of the ALFA-tag-insertion site of DSCAM were described in our previous paper[48]. Because DSCAM has a signal peptide for membrane insertion at the N-terminus and a PDZ binding motif at the C-terminus, terminal fusion of the artificial tag may inhibit the DSCAM functions. To determine the tag-insertion site,

the amino acid sequence for the mouse DSCAM protein was compared with that of DSCAM from other species using the web-based protein structure prediction tool Jpred 4 (http://www.compbio.dundee.ac.uk/jpred/) to find regions with low stringencies. The selected region was analysed using the web-based CRISPR design tool CRISPOR (http://crispor.org) in order to choose the guide RNA sequence with minimised off-target effects. ALFA-tag was inserted into the cytoplasmic region to obtain the *Dscam*^ALFA KI mouse line, which was confirmed to show no changes in total DSCAM expression levels. Mice homozygous for *Dscam*^ALFA were viable and fertile, exhibiting normal development with regard to body size and weight, and did not present any abnormalities throughout the study period. Therefore, this carefully evaluated line was used in this study.

### Generation of Dscam^ALFA knock-in allele by CRISPR/Cas9-mediated genome editing

In our previous study, the insertion of two tandem copies of PA-tag into the Dscam cytoplasmic region did not affect *Dscam* expression levels or cause any abnormalities during the developmental stage in the *Dscam*^PA knock-in mice[48]. To generate a new *Dscam*^ALFA knock-in allele, three tandem copies of ALFA-tag, a recently designed artificial peptide (SRLEEELRRRLTE), were inserted into the same site within the cytoplasmic region as in the *Dscam*^PA knock-in allele. ALFA-tag forms a stable α-helix that can be specifically detected by a high-affinity nanobody[49]. To minimise any potential influence from neighbouring secondary structures, each ALFA-tag peptide was framed by prolines (P), and a linker peptide (AGA) was placed between the three copies of ALFA-tag, resulting in the insertion of 51 amino acid long P-(ALFA-tag)-PAGAP-(ALFA-tag)-PAGAP-(ALFA-tag)-P peptide as a whole. The previously used CRISPR guide RNA sequence[48] was again employed in this study. Two parts of CRISPR guide RNA, Dscam-crRNA (CRISPR RNA) [5′-UUGUUAAACCGGGGCGCACCguuuuagagcuaugcu-3′, with modifications to block exonucleases' attacks] and tracrRNA (trans-activating CRISPR RNA) [5′- AGCAUAGCAAGUUAAAAUAAGGCUAGUCCGUU AUCAACUUGAAAAAGUGGCACCGAGUCGGUGCUUU-3′, with modifications to block endonucleases'/exonucleases' attacks] were chemically synthesised by Integrated DNA Technologies (IDT). Alt-R S.p. HiFi Cas9 Nuclease V3 was purchased from IDT. The single-stranded DNA (ssDNA) donor was subsequently designed to contain three tandem copies of ALFA-tag sequences flanked by 23-24 nucleotides homology arms on both sides. It was chemically synthesised by IDT in the form of Alt-R HDR Donor Oligo, which had IDT's proprietary modifications on both sides to enhance knock-in efficiencies. In detail, the ssDNA donor consisted of the left homology arm [5′-GAC TTCTTGTTAAACCGGGGCGCA-3′], the ALFA-tag sequences [5′-CCG AGCCGCCTGGAAGAAGAACTGCGCCGCCGCCTGACCGAACCCGCTG GAGCACCCTCCCGGCTCGAGGAAGAGCTCCGCCGGCGGCTCACAGA GCCAGCTGGAGCACCGTCCAGACTCGAAGAGGAGCTGAGACGGAGG CTCACTGAACCC-3′], and the right homology arm [5′-CCAGGCAC CAGCAGGGACCTGAG-3′]. The guide RNAs, Cas9 nuclease, and the ssDNA donor were electroporated into B6C3F1 mouse zygotes following the standard protocol[93]. By performing PCR using a pair of primers, F [5′- GTGCCTCTCTGAAGTGCACTCTAATG-3′] and R [5′- TGC ACCCTCTCGCTGAGGTAAG-3′], and the subsequent Sanger sequencing analyses, two founders were confirmed to carry the correct knock-in allele. These founders were crossed with C57BL/6J mice to obtain heterozygous F1 progenies, which were again verified to harbour the designed knock-in by Sanger sequencing. Homozygous mice after F2 generations were then used for western blotting and immunohistochemical analyses.

### Nissl staining

Fixed tissues were cryosectioned at 10 μm thickness, delipidated with the mixture (Chloroform: Ether: Et-OH = 7: 1: 1) for 7 min on slides, and were dehydrated stepwise using EtOH (100%, 70%, 50%) and DW for 1 min. The tissues were stained with Cresyl violet reagent (0.1% Cresyl violet, 1% Acetic acid) for 1 min and washed with DW. Second dehydration was processed stepwise using EtOH (50%, 70%) for 5 min, 100% EtOH for 5 min (twice), and Xylene for 5 min (twice).

### Antibodies, immunohistochemistry, and immunocytochemistry

Tissues were fixed with 4% paraformaldehyde (PFA) in PBS, cryoprotected by overnight immersion in 30% sucrose in PBS, embedded in optimum cutting temperature (Tissue-Tek O.C.T. compound; Sakura Finetek, Tokyo, Japan), and cryosectioned at 20 or 40 μm for fluorescence observation of neuronal morphology (CM3050 S; Leica, Wetzlar, Germany). Sagittal sections of the cerebellum were used for the experiments. Most of the data were obtained from the sulcus of lobules 6. The sections were soaked in PBS for 5 min, pre-incubated in blocking buffer containing 0.1% Triton X-100, 1% normal donkey serum (Millipore, Billerica, MA, USA), or 1% BSA at 25 °C for 1 h, and subsequently immunolabeled using the following primary antibodies in blocking buffer at 4 °C overnight. The following primary antibodies were used: chick anti-GFP (1:500; ab13970; Abcam, Cambridge, UK), goat anti-Calbindin (1:500; AB_2571569; Frontier Institute, Hokkaido, Japan), guinea pig anti-vGluT2 (1:500; AB_2571621; Frontier Institute), goat anti-vGluT2 (1:500; AB_2571620; Frontier Institute), guinea pig anti-vGluT1 (1:500; AB5905; Millipore), guinea pig anti-VGAT (1:500; AB_2571624; Frontier Institute), rat anti-GFAP (1:500; 13-0300; Thermo Fisher Scientific, Waltham, MA, USA), goat anti-GLAST (1:500; AB_2571716; Frontier Institute), and rabbit anti-PKCγ (1:500; AB_2571824; Frontier Institute).

For the detection of endogenous DSCAM in WT mice, the following primary antibodies were tested: rabbit anti-DSCAM (1:100–10000; LS-B5787-50; LifeSpan Biosystems, Minneapolis, MN, USA), mouse anti-DSCAM (1:100–10000; MAB2603; Millipore), rabbit anti-DSCAM (1:100–10000; NBP2-30716; Novus Biologicals, Centennial CO, USA), rabbit anti-DSCAM (1:100–10000; ORB156648; Biorbyt, Cambridge, UK), rabbit anti-DSCAM (1:100–10000; sc79437; Santa Cruz Biotechnology, Santa Cruz, CA, USA), rabbit anti-DSCAM (1:100–10000; HPA019324; Sigma-Aldrich, St. Louis, MO, USA), goat anti-DSCAM (1:100–10000; AF3666; R&D Systems, Minneapolis, MN, USA), and rabbit anti-DSCAM antibody produced in our laboratory (antigen: 1880-2008 amino acids).

Specimens were subsequently rinsed with PBS and incubated with secondary antibodies of AffiPure F(ab′)₂ Fragment conjugated with Alexa Fluor 488, 568, 594, or 647 (1:1,000; Jackson ImmunoResearch Laboratories, Baltimore Pile, PA, USA) or Alexa Fluor® (1:1,000; Abcam) and DAPI (5 μg/mL; Invitrogen, Carlsbad, CA, USA) in blocking buffer in PBS at 25 °C for 2 h.

To detect the DSCAM-ALFA in *Dscam*^ALFA/ALFA mice, the cerebellar slices were incubated with pepsin solution (ab64201, Abcam) at 37 °C for 10 min. The slices were washed with PBS two times and incubated with the primary antibodies listed above. The nanobody fused with Alexa647 (FluoTag-X2 anti-ALFA conjugated Alexa647, N1502-AF647-L, NanoTag Biotechnologies, Gottingen, Germany) was incubated with secondary antibodies.

Fluorescence imaging was performed using a Zeiss LSM 780 confocal microscope system (Carl Zeiss, Oberkochen, Germany) and ZEN 2009 software (Carl Zeiss) or spinning disk confocal superresolution microscope (SpinSR10, Olympus Corporation, Tokyo, Japan) and cellSens Dimension (Ver 3.1.1, Olympus Corporation).

To detect DSCAM-ALFA at both low and high magnifications, a Leica Stellaris 5 laser scanning confocal microscope (Leica, Wetzlar, Germany) was used and images were obtained with the LASX software (Leica). For high magnification, the Lightning 3D deconvolution method was used for image acquisition.

Quantification of the fluorescence intensity of immunolabeled cells was performed using the "Measure" and "Plot Profile" functions of ImageJ v1.46r. The number of cells, dendritic height, and the height

and number of vGluT2 puncta were measured using Fiji/ImageJ 2.3.0/1.53. For measurements of the height and puncta number of vGluT2, puncta with sizes greater than 0.5 μm$^2$ and smaller than 13 μm$^2$ were defined as vGluT2-positive CF presynapses whereas puncta smaller than 0.5 μm$^2$ or located outside the dendritic shafts were not[11]. The configured puncta size was automatically counted using "Analyze Particles". The density of vGluT2 puncta was automatically measured and calculated using "Trainable Weka segmentation", "Auto Threshold", and "Analyze Particles" modules in Fiji/ImageJ with slight modifications.

## Brain lysate preparation

To investigate the temporal expression profile of DSCAM in the cerebellum (Fig. 1c), the whole cerebellum from P0 to P28 was isolated in PBS and homogenised in homogenisation buffer (20 mM Tris, 1 mM EDTA, 1 mM DTT, 150 mM NaCl, 100 mM sucrose, protease inhibitor cocktail [cOmplete; Roche, Basel, Switzerland]; pH 7.4, 4 °C). Samples were clarified by centrifugation at 20,000 g for 10 min at 4 °C. The resultant pellets were mixed with lysis buffer (10 mM Tris, 1 mM EDTA, 150 mM NaCl, 1% NP-40, protease inhibitor cocktail; pH 7.4, 4 °C), sonicated for 15 s on ice, and boiled in SDS-sample buffer for 5 min at 95 °C.

## Synaptic fraction preparation

To confirm DSCAM localisation at the synaptic fraction, P21 BALBc mouse cerebella were homogenised in Syn-PER synaptic protein extraction reagent (#87793, Thermo Fisher Scientific). All steps were performed at 4 °C or on ice, and all buffers contained a protease inhibitor cocktail (cOmplete; Roche). The samples were centrifuged at 1200 · g for 10 min. The collected supernatant was centrifuged at 15,000 · g for 20 min, and the resultant pellets were used for the following assays. For the co-immunoprecipitation assay using the synaptosomal fraction, the resultant pellets were resuspended in ULTRARIPA Kit for Lipid Raft B-buffer (F015; BioDynamics Laboratory, Tokyo, Japan). For the western blot analyses, the resultant pellets were resuspended in the Syn-PER synaptic protein extraction reagent and boiled in SDS-sample buffer for 5 min at 95 °C.

## Western blot analyses

Samples were subjected to SDS-PAGE using a gradient or 12% acrylamide gel. Proteins transferred onto a PVDF membrane were immunoblotted with the following primary antibodies: rabbit anti-DSCAM (1:1000, HPA019324; Sigma-Aldrich), goat anti-RORα (1:1000, sc-6062; Santa Cruz Biotechnology), rabbit anti-β-actin (1:1000, sc-47778; Santa Cruz Biotechnology), rabbit anti-EAAT1/GLAST (1:500; ab416; Abcam), rabbit anti-GluD2 (1:500, AB_2571600; Frontier Institute), mouse anti-PSD95 (1:1000, MA1-045; Invitrogen), rabbit anti-Cul3 (1:1000, ab72187; Abcam), rabbit anti-GFP (1:1000, #598; MBL, Woburn, MA, USA), rabbit anti-Flag (1:1000, SAB4301135; Sigma-Aldrich), rabbit anti-pMARCKS (Ser152/156) (1:1000, #07-1238; Millipore), and mouse anti-MARCKS (1:1,000, WH0004082M6; Sigma-Aldrich). The peroxidase-coupled antibodies anti-mouse IgG (1:1000, PI-2000; Vector Laboratories, Burlingame, CA, USA), anti-rabbit IgG (1:1000, PI-1000; Vector Laboratories), and anti-goat IgG (1:1000, PI-9500; Vector Laboratories) were used as secondary antibodies for immunoblotting. Signals were detected using a cooled CCD camera (LAS-4000 mini; Fujifilm, Kanagawa, Japan).

## In utero electroporation

Pregnant E12.5 mice were anesthetised by intraperitoneal administration of dexmedetomidine (0.3 mg/kg; Zenoaq, Tokyo, Japan), midazolam (2 mg/kg; Astellas Pharma, Tokyo, Japan), and butorphanol tartrate (2.5 mg/kg; Meiji Seika Pharma, Tokyo, Japan) on a heating pad. The uterine horns were exposed, and ~1 μL plasmids mixed with Fast Green (Sigma-Aldrich) in TE buffer were manually injected into the developing aqueduct using a pulled glass micropipette (G-1.0; Narishige, Tokyo, Japan). Concentrations were as follows: 0.25 μg/μL pCAG−hyPBase, 0.25 μg/μL pPB-CAG-DsRed2, and 1.5 μg/μL pCAG−Dscam−mEGFP. Electric pulses (33V, four pulses; 30 ms on, 970 ms off) were delivered across the heads of the embryos targeting the dorsal-medial part of the cerebellum with forceps electrodes (CUY650P5; Nepa Gene, Ichikawa City, Japan) connected to electroporator (CUY21E; Nepa Gene). After electroporation, uteri were placed back into the abdominal cavity, allowing embryos to continue developing. All surgical procedures were completed within 30 min, after which the mice were recovered on a heating pad for 30 min. P15 mouse brains were fixed and subjected to immunohistochemical analysis.

## Immunoprecipitation assay

To identify the DSCAM−GLAST interaction, the synaptosomal fraction resuspended in ULTRARIPA Kit for Lipid Raft B-buffer (BioDynamics Laboratory) was sonicated for 15 min on ice and clarified by centrifugation at 200,000 · g for 10 min at 4 °C. The collected supernatant was incubated with protein A Sepharose (GE Healthcare, Chicago, IL, USA) and 10 μg control rabbit IgG, rotated for 30 min at 4 °C, and centrifuged at 12,000 g for 1 min at 4 °C. The resultant supernatant was incubated with 1 μg rabbit anti-DSCAM (HPA019324; Sigma-Aldrich), anti-EAAT1 (GLAST, ab416; Abcam) antibodies or control rabbit IgG (011-000-003; Jackson ImmunoResearch Laboratories), and rotated for 2 h at 4 °C. The immunocomplexes were then precipitated with protein G Sepharose 4 B (GE Healthcare) for 1 h at 4 °C. After three washes with PBST buffer (0.05% Tween20 and protease inhibitor cocktail [cOmplete] in PBS; pH 7.4, 4 °C), the obtained eluates were boiled in SDS-sample buffer for 5 min at 95 °C and then subjected to SDS-PAGE and western blotting.

For the immunoprecipitation of endogenous DSCAM-ALFA, the whole brains from P10 and P11 WT (*Dscam*$^{+/+}$) or *Dscam*$^{ALFA/ALFA}$ mice were suspended in RIPA buffer and sonicated for 15 min on ice. The samples were then clarified by centrifugation at 200,000 xg for 10 min at 4 °C. After removing the supernatants, the collected pellets were resuspended with ULTRARIPA Kit for Lipid Raft B-buffer. The resuspended brain lysates were incubated using the Dynabeads Protein G Immunoprecipitation Kit (10007D; ThermoFisher Scientific) and rotated for 30 min at 4 °C, followed by centrifugation at 12,000 x g for 1 min at 4 °C. The resultant supernatant was incubated with 1 μg rabbit anti-ALFA (N1581; NanoTag Biotechnologies), rabbit anti-EAAT1/GLAST (#5684; Cell signaling technologies) antibodies or control rabbit IgG (PM035; MBL, Tokyo, Japan), and rotated for 2 h at 4 °C. The immunocomplexes were then precipitated using the Dynabeads Protein G Immunoprecipitation Kit for 1 h at 4 °C.

## RNA sequencing and data analysis

Total RNA was extracted from the cerebellum tissues of five WT and five *Dscam*$^{del17/del17}$ mice at P21 using the RNeasy Plus Universal Kit (Qiagen, Hilden, Germany). RNA quantity and quality were assayed using a NanoDrop and Qubit Fluorometer (Thermo Fisher Scientific). Sequencing libraries were prepared using the NEBNext Ultra Directional RNA Library Prep Kit for directional libraries (New England BioLabs, Tokyo, Japan). RNA-seq libraries were sequenced using Illumina NovaSeq platforms. Adapter and low-quality sequences were trimmed using Trimmomatic (version 0.36)[94], and the trimmed reads were aligned to the reference mouse genome (GRCm38/mm10) using HISAT2 (version 2.2.0)[95]. Genome-wide expression levels were measured as a unit of transcripts per kilobase million (TPM) using StringTie (version 1.3.6)[96] and the number of reads was counted per gene per sample using htseq-count within HTSeq (version 0.9.1)[97]. Differentially expressed genes were identified using DESeq2 (version 1.8.2)[98]. For gene ontology analysis, DAVID Bioinformatics Resources (version 6.8) was used (National Institute of Allergy and Infectious Diseases, National Institutes of Health; https://david.ncifcrf.gov).

## Horizontal optokinetic response

Pcp2[Cre]-cKO mice at postnatal 8 to 10 weeks were anesthetised with an intraperitoneal injection of ketamine/xylazine (80/20 mg/kg; Sigma-Aldrich), and a 1-cm flat-head screw was attached to the skull using synthetic resin cement (Super-Bond; Sun Medical, Shiga, Japan). After several days, the mice were positioned at a table, their heads were fixed with a screw, and their bodies were loosely restrained with plastic cylinders. Experiments examining the horizontal optokinetic response (OKR) employed the black and white checked-pattern screen that oscillates sinusoidally around the animal (screen height, 55 cm from the eye of the mouse: check size, 2-cm square) by 15° (peak-to-peak) at 0.33 Hz. The OKR was observed for 1 h. Evoked eye movements without blinks or saccades were averaged over 10 cycles, and the average amplitude was calculated using modified Fourier analysis as previously described[99]. The gain of eye movement was defined as the ratio of the peak-to-peak amplitude of the eye movement to the screen oscillation.

## Electrophysiology

Parasagittal cerebellar slices (200 µm thick) were prepared from *Dscam*[del17/del17] mice (P25–35), as described previously[100]. Whole-cell patch-clamp recordings were made from visually identified Purkinje cells using a 60· water-immersion objective attached to an upright microscope (BX51WI; Olympus, Tokyo, Japan) at 25 °C. The resistance of the patch pipettes was 1.5 – 3 MΩ when filled with an intracellular solution composed of 150 mM Cs-gluconate, 10 mM HEPES, 4 mM MgCl₂, 4 mM Na₂ATP, 1 mM Na₂GTP, 0.4 mM EGTA, and 5 mM lidocaine N-ethyl bromide (QX-314) (pH 7.25, 295 mOsm/kg). The solution used for slice storage and recording consisted of the following: 125 mM NaCl, 2.5 mM KCl, 2 mM CaCl₂, 1 mM MgCl₂, 1.25 mM NaH₂PO₄, 26 mM NaHCO₃, and 10 mM D-glucose, which was bubbled continuously with a mixture of 95% O₂ and 5% CO₂. Picrotoxin (100 µM; Sigma) was always present in saline to block inhibitory inputs.

To evoke EPSCs derived from the CF and PF inputs onto the Purkinje cells, square pulses were applied through a stimulating electrode placed on the granular layer and the molecular layer (-50 µm away from the pial surface). Selective stimulation of the CFs and PFs was confirmed by paired-pulse depression and paired-pulse facilitation, respectively, of EPSC amplitudes at 50-ms interstimulus intervals. During CF-EPSC recordings, 500 nM 2,3-dioxo-6-nitro-1,2,3,4-tetrahtdrobenzo[f]quinoxaline-7-sulfonamide disodium salt (NBQX; an AMPAR/kainate receptor antagonist; Tocris Bioscience, Bristol, UK) was added to the artificial cerebrospinal fluid to avoid the voltage-clamp error caused by the large EPSC responses[101]. Current responses were recorded using an Axopatch 200 B amplifier (Axon Instruments, Sunnyville, CA, USA), and the pCLAMP system (v9.2; Axon Instruments) was used for data acquisition and analysis. The signals were filtered at 1 kHz and digitised at 4 kHz. In electrophysiological analyses using mutants or with drugs, the data were not normally distributed. Therefore, Mann-Witeny U test was used for statistical analysis, because this test can be adapted to non-normally distributed data groups and deal with relatively small sample sizes.

## Electron microscopy

Tissues were fixed with 4% PFA/ 0.1% glutaraldehyde (GLA) or 2% PFA/ 2% GLA in 100 mM cacodylate buffer. After embedding in immunogold, 100-µm-thick sections were cut with a microtome (Leica) and the sections were incubated with goat anti-GLAST antibody diluted with 1% BSA and 0.004% saponin in PBS overnight. Anti-goat IgG linked to 1.4 nm gold nanoparticles was incubated for 2 h, and gold nanoparticles were intensified using a silver enhancement kit (R-Gent SE-EM; Aurion, Wageningen, Netherlands). Sections were fixed with 1% osmium tetroxide (OsO₄) in cacodylate buffer for 20 min and washed six times with H₂O for 10 min. Dehydration was performed stepwise using EtOH (50, 70, 80, 90, and 95%) for 10 min, 100% EtOH for 15 min twice, QY-1:100% EtOH (1:1) for 10 min, and QY-1 for 15 min twice. After

dehydration, the samples were embedded in Epon for 2–3 days at 62 °C. Sections (70 nm thick) were cut using an ultramicrotome (Ultracut; Leica), mounted on grids, stained with 2% uranyl acetate for 30 min, and examined with transmission electron microscopy (TEM, JEM-3200FS; JEOL, Tokyo, Japan). For GLAST immunoelectron microscopy, images in which gold nanoparticles were not observed on the Bergmann glial membrane and non-specifically scattered in nerve cells were excluded as observation targets. CF and PF synapses were distinguished by the previously reported morphological features[102]. Small terminals connecting to single spines originate from PF synapses, whereas large terminals with multiple contacts originate from CF synapses. In our experimental conditions, the immunoelectron microscopy images only provided data for less than 10 synapses per image. Although a large number of images were acquired and analysed, a sufficient number of CF synapses was not obtained for statistical evaluation. Therefore, the statistical analysis of GLAST-nanoparticle localisation was performed only in PF synapses. Because of silver enhancement, signals varied in size and were not often round in shape. In addition, the silver enhancement caused an extremely fine-grained background. We therefore only counted signals on the plasma membrane at 10–30 nm as significant signals.

## Cell culture, transfection, and immunoprecipitation assay

COS-7 cells were cultured in Dulbecco's modified Eagle's medium (DMEM) supplemented with 10% foetal bovine serum (FBS). Transfection was performed using Lipofectamine LTX (Invitrogen), according to the manufacturer's instructions. After transfection for 24 h, cells were scraped in lysis buffer (20 mM Tris, 150 mM NaCl, 1 mM EDTA, 0.1% NP-40, and protease inhibitor cocktail [Roche]; pH 7.4, 4 °C), sonicated for 15 s on ice, and then clarified by centrifugation at 20,000 · g for 10 min at 4 °C. The soluble supernatants were incubated with 1 µg mouse IgG or rabbit anti-FLAG (M2; Sigma-Aldrich) for 2 h at 4 °C. The immunocomplexes were then precipitated with protein A Sepharose 4 B for 1 h at 4 °C. After three washes with lysis buffer, the eluates were subjected to SDS-PAGE and immunoblotting.

## RNA extraction and RT-PCR

Mouse brain tissue was disrupted with Qiazol Lysis Reagent (#79306; Qiagen) and Biomasher (#320102; Nippi, Tokyo, Japan), and mRNA was extracted using the RNeasy Plus Universal Midi Kit (#73442; Qiagen). Reverse transcription of the extracted mRNA was performed using the ReverTra Ace qPCR RT Master Mix (FSQ-201; Toyobo, Osaka, Japan). The primer sequences are shown below. Dscam Exon1 (Forward: 5′-CCCTCCTGTGACTCTCAGATG-3′, Reverse: 5′-CCCGTAAAACAGCCTTGATGTG-3′), Dscam Exon2-3 (Forward: 5′-ATCCTGGACGGGTTTGACCA-3′, Reverse: 5′-GCGGCTGTTCTCAATGAGC-3′), Dscam Exon15 (Forward: 5′-GTACCCAGCTACCCTCCTGA-3′, Reverse: 5′-CTCCTTGGACAGTGTGGACC-3′), Dscam Exon21 (Forward: 5′-TCCTCCATCACCCTCTCCTG-3′, Reverse: 5′-TTTCCCCAGGGTTTTGGCTT-3′), and β-actin (Forward: 5′-GGCTGTATTCCCCTCCATCG-3′, Reverse: 5′-CTCCTTGGACAGTGTGGACC-3′). PowerUp SYBR Green Master Mix (A25742; Applied Biosystems) was used for PCR, and amplification and detection were performed using LightCycler96 (Roche). The relative amounts of mRNA were calculated using the ΔΔCt-method. To briefly describe the calculation method, the Ct value of *Dscam* in each sample was ΔCt value-corrected by the Ct value of β-actin. ΔCt values were compared relative to each other (ΔΔCt value).

## Riluzole administration

Riluzole (#557324; Sigma-Aldrich) was dissolved in 10% (2-Hydroxypropyl)-β-cyclodextrin solution (#18847-64; Nacalai Tesque, Japan)[103,104], and then diluted with normal saline to a final experimental concentration (1.3 mg/ml). Riluzole (13 mg/kg/day)[70,71] or vehicle (saline solution) (Otsuka saline solution; Otsuka Pharmaceutical Factory, Japan) was administered intraperitoneally from P15 to P29 once a day.

## Statistical analysis

Statistical analyses were performed using Prism v7.0 (GraphPad Software, La Jolla, CA, USA) or R Studio v3.0 (R-Tools Technology, Boston, MA, USA). All data are expressed as the mean ± SEM unless stated otherwise. Binary continuous variables were compared using an unpaired two-tailed $t$-test, while binary continuous variables with different numbers were compared using the Mann-Whitney test. Multiple continuous variables were compared using a two-way analysis of variance and Tukey's multiple comparison test. Statistical significance was set at $P < 0.05$. No randomisation of samples was performed. No blinding was performed. The number of samples or cells examined in each analysis is indicated in the figure legends.

## Reporting summary

Further information on research design is available in the Nature Portfolio Reporting Summary linked to this article.

## Data availability

All data needed to evaluate the conclusions in the paper are presented in the paper and/or Supplementary Information. Additional data are available from the authors upon reasonable request. RNA-seq data were deposited in the GEO database with the accession number GSE190010. Source data are provided with this paper.

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

## Acknowledgements

We thank Dr. Ryohei Yasuda (Max Plank Florida Institute for Neuroscience) for pCAG–hyPBase and pPB-CAG-DsRed2. We also thank Dr. Hiroshi Kawabe (Gumma University) for fruitful discussions, and Mrs. Ikuko Hasegawa for technical assistance. This work was supported by Grants-in-Aid for Scientific Research, KAKENHI (Grant #22H02730 to M.H.; Grant #17K07071 and #21K06416 to N.A.; Grant #17K07126 and #20K06886 to S.T.; Grant #16H06531 to Y.G., #21H05243 to T.S.); JST (JPMJCR21P1; JPMJMS2292 to T.S.); AMED (JP23wm0425005h0003 M.H. and S.T.); an Intramural Research Grant of NCNP (Grants 3-9, 4-5, 4-6 for M.H.); Japan Health Research Promotion Bureau (JH) under Research Fund (Grant 2020-B-07 for M.H); Tokumori Yasumoto Memorial Trust to M.H.; the Naito Foundation, the Takeda Foundation, and the Daiichi Sankyo Foundation of Life Science, The Sumitomo Foundation, and the JSPS Research Fellow DC1 (Grant #JP17J07118 to K.D.).

## Author contributions

K.D., N.A., S.Taya, and M.H. supervised the project. K.D., N.A., and M.H. designed the study and analysed the data. K.D., N.A., S.M., T.A., A.T., K.Hizawa, H.U., K.Hori, H.Y., C.U., M.I., S.Tatsumoto, M.W., W.K., K.S., Y.I., T.I., D.M., and Y.G. performed the experiments. K.D., N.A., and M.H. wrote the manuscript. K.D., N.A., S.Taya, T.M., N.I., T.S., J.N., M.S., Y.G., Y.K., K.Y., K.K., S.K., M.Y., and M.H. discussed the experimental results and commented on the manuscript.

## Competing interests

The authors declare no competing interests.

## Additional information

[1]Department of Biochemistry and Cellular Biology, National Institute of Neuroscience, NCNP, Tokyo 187-8502, Japan. [2]Department of Neuropharmacology, Interdisciplinary Graduate School of Medicine, University of Yamanashi, Chuo, Yamanashi 409-3898, Japan. [3]Laboratory for Glia-Neuron Circuit Dynamics, RIKEN Center for Brain Science, Wako, Saitama 351-0198, Japan. [4]Department of Pharmacology, Graduate School of Pharmaceutical Sciences, Tohoku University, 980-8578 Tohoku, Japan. [5]Department of Physiology, Keio University School of Medicine, Tokyo 160-8582, Japan. [6]Department of System Pathology for Neurological Disorders, Brain Research Institute, Niigata University, Niigata 951-8585, Japan. [7]Division of Behavioural Neuropharmacology, International Center for Brain Science, Fujita Health University, Toyoake, Aichi 470-1192, Japan. [8]Department of Health Sciences, School of Medicine, Hokkaido University, Sapporo, Hokkaido 060-0812, Japan. [9]Cognitive Genomics Research Group, Exploratory Research Center on Life and Living Systems, National Institutes of Natural Sciences, Okazaki, Aichi 444-8585, Japan. [10]Department of Biomolecular Science, Faculty of Science, Toho University, Funabashi, Saitama 274-8510, Japan. [11]Department of Ultrastructural Research, National Institute of Neuroscience, NCNP, Tokyo 187-8502, Japan. [12]Department of Neurodevelopmental Disorder Genetics, Nagoya City University Graduate School of Medicine, Nagoya, Aichi 467-8601, Japan. [13]Department of Life Science Frontiers, Center for iPS cell Research and Application (CiRA), Kyoto University, Kyoto 606-8507, Japan. [14]Department of System Neuroscience, National Institute for Physiological Sciences, National Institutes of Natural Sciences, Okazaki, Aichi 444-8585, Japan. [15]Graduate School of Information Science, University of Hyogo, Kobe, Hyogo 650-0047, Japan. [16]Division of Cell Biology, International Center for Brain Science, Fujita Health University, Toyoake, Aichi 470-1192, Japan. [17]Department of Anatomy, Faculty of Medicine, Hokkaido University, Sapporo, Hokkaido 060-8638, Japan. [18]The University of Texas at Austin, Austin, Texas 78712-0805, USA. ✉e-mail: nariko.arimura.a2@tohoku.ac.jp; hoshino@ncnp.go.jp

