## [Peer Review File · Nature Communications]

Neuronal DSCAM regulates the peri-synaptic localization of GLAST in Bergmann glia for functional synapse formationREVIEWER COMMENTS

Reviewer #1 (Remarks to the Author):

The study by Dewa et al. addresses the function of DSCAM receptors for function and localization of the glutamate transporter GLAST in the context of synapse formation in the cerebellum. The authors provide genetic evidence that cell specific knock-out of DSCAM in cerebellar Purkinje neurons results in synaptogenesis defects, that – at least in part – is due to altered GLAST localization/function. This model is supported by phenotypic similarities of DSCAM and GLAST mutant neurons as well as rescue by (pharmacologically) reducing free glutamate. This set of results is experimentally well supported. The analysis, however, how the synaptic defects arise, how the molecular function of DSCAM give rise to the cellular/physiological defect remain unclear and preliminary. The question of whether, how, or when DSCAM receptors control GLAST localization remains uncharacterized.

Detailed critique:

It is essential that the localization of the endogenous DSCAM receptor is determined precisely. Otherwise no cellular/mechanistic model in the context of GLAST localization/restriction can be established. Over-expression of tagged receptors without temporal or quantitative control is not appropriate. The authors vaguely allude to problems with antibodies but it is unclear what the problems really are and/or what exact antibodies were tested. If indeed antibodies could not be used, then tagging of endogenous gene would be the method of choice, which ideally enables a cell-type specific assessment of protein localization (e.g. Fang et al 2021, eLife 10:e65202).

For cellular/molecular models this study needs a better temporal resolution: How do the phenotypes arise during development.

Page 7 line 147. It is essential to provide detailed information how the fusion-protein was generated and whether potential impairment of function/localization can be excluded.

For DSCAM knock-out experiments it is important to describe and summarize what phenotypic traits have actually been examined (examining cerebellar development). Given the known functions, it would seem important to determine potential self-avoidance defects that may indirectly impact on synaptogenesis.

Page 7 line 163: seems to be the wrong citation, as this paper (Fuerst et al.) examines the role of DSCAM only in the visual system.

Page 7 line 167: phenotype needs to be described precisely. What is the cellular phenotype? Are there less cells, more apoptosis, etc. Quantification would be very helpful.

Page 4 line 72/73: poor sentence - the term localization rather than recognition should be used

Page 14 line 343: "intracellular heterophilic association". This is incorrect, presumably, the authors wanted to use the term "intercellular". That section needs to be clarified as it actually refers to a proposed mechanistic model (not yet supported by data).

Throughout the entire text, I do not understand what the authors refer to when they use the term "relocate". Are they referring to dendritic targeting?

Page 5, Line 91-94. Sentence makes no sense. What is "unbalanced localization"?

Reviewer #2 (Remarks to the Author):

The manuscript submitted by Dewa et al describes the role of the down syndrome cell adhesion molecule (DSCAM) expressed in Purkinje cells in the proper localization of the glial glutamate transporter GLAST presenting Bergmann glia cells. The results presented in this contribution are important not only to understand basic neuron/glia interactions in terms of the regulation of tripartite synapses, but also to gain insight into the developmental processes of climbing fiber-Purkinje synapses. It is a well-written easy to follow contribution that requires minor modifications before it can be accepted for publication.

1. Although the authors tend to focus on the climbing fiber-Purkinje cell synapse (CF), mostly in terms of its formation, the fact that the absence of DSCAM impairs the parallel fibers-Purkinje cell synapse (PF) might prove to be an important finding taking into consideration that it is thus synapse (PF) the most abundant glutamatergic synapse in the whole brain. In this context, it might be important to discuss in a more deep detail, the fact that DSCAM is required for motor learning but not for motor coordination.

2. Taking into consideration that Bergmann glia express very low levels of the other glial glutamate transporter Glt-1, it is surprising that disrupted GLAST localization in the DSCAM-depleted animals the most relevant finding is the failure to establish proper CF synapses. One would expect a severe neurodegenerative scenario due to the excessive glutamate present into only in the synaptic cleft but also the spilled over neurotransmitter. The authors are prompted to provide an explanation in the Discussion.

3. As the authors correctly point out, DSCAM binds to Glt-1. Being Glt-1 roughly 1% of total brain protein the authors might consider to speculate if DSCAM-Glt-1 is also responsible of a proper Got-1 localization.

4. One of the authors main conclusions is that augmented free glutamate is responsible of the impairment of CF synaptogenesis, it is important that the authors elaborate in a plausible mechanism (Page 15, second paragraph).

5. As depicted in Extended Data Fig 10 panel C, DSCAM is needed for the proper GLAST localization and function. This is not a minor issue and the scenarios clearly more complex than the one depicted in the referred figure. In fact, this is a major finding, please discuss this in the context of cerebellar function, since failure of a proper glutamate uptake triggers a severe energetic deficit (see *Neuropharmacology* 192 (2021) 108602 for a review., DOI: 10.3389/fphys.2021.825816).

Reviewer #3 (Remarks to the Author):

This is a very interesting manuscript that indicates the role of Down syndrome cell adhesion molecule (DSCAM) in peri-synaptic localization of Bergmann glia glutamate transporter GLAST. Authors present data that suggest that DSCAM loss and consequently reduced peri-synaptic GLAST lead to impaired parallel fiber to Purkinje cell synaptic signaling and climbing fiber synaptogenesis, with intact basic CF and PF ephys and BG wrapping of PC synapses. Authors have also used riluzole to ameliorate impaired CF translocation. While this is an interesting and significant finding, some of the conclusions are not fully supported by data and additional information is needed for reviewer to better evaluate manuscript.

1. In Fig 4 authors investigate whether GLAST interacts with DSCAM performing CO-IP of endogenous proteins in cerebellar synaptosomes and overexpressed proteins in COS-7 cells. This is an important figure as it demonstrates novel DSCAM interaction with astrocytic protein. Co-IPs are usually done both ways to rigorously show that proteins can precipitate each other. Could authors please show that DSCAM can IP GLAST? Authors also state in Fig legends that this was repeated 3 times (n=3). IP in Fig 4a indicates a very weak interaction (DSCAM band is large in the input and very faint in GLAST-IP) -therefore showing all three independent repeats would be important to support the conclusion. In Fig 4c IP is done with overexpressed proteins in COS7 cells- could authors please indicate which of many bands is specific for the GLAST?

2. In the abstract authors claim that "Dscam-mutant mice mimicked loss-of-function mutations in the astrocytic glutamate transporter GLAST expressed in Bergmann glia." This is not accurate -GLAST loss of function mice have different phenotype from DSCAMdel7/del7 mice described in this manuscript (i.e. impaired glial wrapping of synapses vs no changes here, multiple CF innervation vs no changes here, altered CF-EPSC vs no changes here).

3. Authors next state ". These mice showed impaired glutamate clearance by the delocalization of GLAST, which complexed with the extracellular domain of DSCAM."

This is not fully supported by results.

First, to support impaired glutamate clearance authors should manipulate GLAST (i.e. use blockers of its function such as TBOA and enhancers such as riluzole) and show differential effect on wt and DSCAMdel7/del7 PF-PC synapses.

Second, to support delocalization of GLAST it is critical to show that wrapping of BG processes around synapses is not altered and that GLAST expression is not altered. In Fig 3L authors show cumulative frequency of GLAST signal from the PSD edge. This result is confusing as 100 was reached at 0.4 μ m distance from the nearest PSD edge. Yet based on extended data 8g only approximately 20% of synapses had BG processes that are closer than 0.4 μ m to PSD. Based on that result only 20% synapses had a chance to have GLAST within 0.4 μ m from PSD (and even that if and only if GLAST was localized at the PSD-closest part of BG processes). Could authors please explain this discrepancy? Could authors also show the average distance of GLAST to PSD per mouse?

In extended data Fig 5 authors investigate whether GLAST expression is altered in DSCAMdel7/del7 mice. For WB in extended data fig5b please show all three WT and DSCAMdel7/del7 samples, include loading control (actin, tubulin etc.) and quantify results.

In extended data Fig 5c and d –quantification indicates decrease in GLAST puncta but it is not statistically significant. However large variability in WT data would indicate that larger number of samples than used here is needed to reach statistical significance. Authors should use power formula and SD of their WT results to calculate number of samples needed.

4. Abstract also states that “Riluzole-induced reduction of free glutamate at the synaptic cleft rescued the impairment of CF synapse formation in Purkinje cell-selective Dscam deficient mice”. This is also not supported by data. First riluzole has many different effects and authors have not examined whether riluzole indeed reduces free glutamate at the synaptic cleft. Experiment proposed above (add riluzole/TBOA to ephys to rescue PF ratio) could address whether riluzole affects glutamate clearance in DSCAM del7/del7 mice. Second, as riluzole did not rescue distal CF innervation this should be stated in the abstract, or clarified that riluzole rescued only proximal impairment of CF synapse formation.

5. . Following information should be added:

1. Methods states “No blinding was performed”. Please clarify why blinding was not used in this study.

2. In Methods please add description of VGLU2 puncta quantification.

3. In Figures 2, 3, extended data 4, 7, and 9 please include number of mice used (if possible in the brackets by the mouse genotype as done in the rest of the figures?)

4. In methods authors states that for CF-EPSC recordings AMPA R antagonist was used. Could authors clarify how can then cyclothiazide (CTZ) which reduces desensitization of AMPA R work in such condition?

5. In Discussion there is paragraph titled “CF synapse selective effect by the depletion of Dscam” in which authors discuss that PF-EPSC is affected and not CF-EPSC and speculate on why. Perhaps title should be changed to indicate this.

6. In Methods for synaptosome preparation authors write “The samples were centrifuged at 1,200 X g for 10 min. The collected supernatant was centrifuged at 1,500 X g for 20 min, and the resultant pellets were used for following assays.” Is the second rotation speed (1500) a type (i.e. should it not be larger?)

Reviewer #1 (Remarks to the Author):

Detailed critique:

1-1

It is essential that the localization of the endogenous DSCAM receptor is determined precisely. Otherwise no cellular/mechanistic model in the context of GLAST localization/restriction can be established. Over-expression of tagged receptors without temporal or quantitative control is not appropriate. The authors vaguely allude to problems with antibodies but it is unclear what the problems really are and/or what exact antibodies were tested. If indeed antibodies could not be used, then tagging of endogenous gene would be the method of choice, which ideally enables a cell-type specific assessment of protein localization (e.g. Fang et al 2021, eLife 10:e65202). For cellular/molecular models this study needs a better temporal resolution: How do the phenotypes arise during development.

Response: Thank you very much for your valuable comment. We also recognize that detecting the localization of endogenous DSCAM is very important to support our hypothesis. We have been trying to detect significant DSCAM signals with seven commercially available antibodies and one lab-made antibody in postnatal WT cerebellar slices. We were able to get some signals using all eight antibodies; however, the signals did not disappear in the *Dscam* cKO mice (*En1^{Cre}; Dscam^{flox/flox}*), which was described in Extended Data Figs. 1c, d of the original manuscript and Supplementary Fig. 1c, d of this revised manuscript. We also confirmed that transcription of *Dscam* exons was certainly eliminated in this cKO mice as shown in Extended Data Fig. 1b of the original manuscript and Supplementary Fig. 1b of the revised manuscript. Therefore, we considered these as false positive signals and concluded that none of the eight antibodies could detect the specific DSCAM signal *in vivo*. The results in cerebellar sections have been described on Pages 6-7, lines 141-146 of the ORIGINAL MANUSCRIPT. To provide further information, the following list of antibody information is presented in the Methods section of the revised manuscript.

Pages 24, lines 596–604

“For the detection of endogenous DSCAM, the following primary antibodies were tested: rabbit anti-DSCAM (1:100–10000; LS-B5787-50; LifeSpan Biosystems, Minneapolis, MN, USA), mouse anti-DSCAM (1:100–10000; MAB2603; Millipore), rabbit anti-DSCAM (1:100–10000; NBP2-30716; Novus Biologicals, Centennial CO, USA), rabbit anti-DSCAM (1:100–10000; ORB156648; Biorbyt, Cambridge, UK), rabbit

anti-DSCAM (1:100–10000; sc79437; Santa Cruz Biotechnology, Santa Cruz, CA, USA), rabbit anti-DSCAM (1:100–10000; HPA019324; Sigma-Aldrich, St. Louis, MO, USA), goat anti-DSCAM (1:100–10000; AF3666; R&D Systems, Minneapolis, MN, USA), and rabbit anti-DSCAM antibody produced in our laboratory (antigen: 1880-2008 amino acids).”

Then, we created three different lines of knock-in mice wherein the DSCAM protein tagged with mEGFP-tag, PA-tag, and HA-tag, respectively were expressed. (i) Knock-in mice designed to express DSCAM with mEGFP inserted in the extracellular domain (FnIII domain) did not express fused DSCAM because of unknown reasons. This was described in our previous paper. [Arimura *et al.*, *Sci Adv*, 2020, 6(36), in Supplementary Methods]. (ii) Therefore, we instead created a knock-in mouse with a tandem PA tag in the intracellular domain immediately flanking the PDZ binding motif (Arimura *et al.*, *Science Advances*, 2020, in Supplementary Fig. S3). Fortunately, this mouse line was useful to visualize the endogenous localization of DSCAM in the embryonic midbrain at E16.5. The PA-tag antibody produced good signals in the E16.5 midbrain of KI mice (*Dscam*^{PA/+}), while this was not observed in the WT mice [Arimura *et al.*, *Sci Adv*, 2020, 6(36), in Fig. 1C & D]. However, this line was not applicable to the postnatal cerebellum because the anti-PA antibody produced very strong background signals in the postnatal cerebellum (this has not been reported anywhere). (iii) Next, we have created a knock-in mouse line designed to express DSCAM protein tagged with tandem HA (Supplementary Fig. 2 of the revised manuscript). However, the signals with anti-HA antibodies were very weak in the postnatal cerebellar slices of the KI mice (*Dscam*^{HA/HA}). For these reasons, (i)-(iii), we decided to analyze the localization of exogenously expressed DSCAM-EGFP fusion protein in the cerebellum, as stated in the original manuscript (Fig. 1d,e of original manuscript and this revised manuscript).

However, considering the reviewer’s suggestion, we again tried to detect the endogenous DSCAM-HA protein in primary cultured neurons from the KI mice (*Dscam*^{HA/HA}). Unfortunately, in the cultured primary neurons from the cerebellum, we detected strong background signals with the anti-HA-tag antibody in WT and KI (*Dscam*^{HA/HA}) mice, as was shown in Supplementary Fig. 2f of the revised manuscript. However, in primary cultured neurons from the hippocampus, we detected significant signals in the KI (*Dscam*^{HA/HA}), which were hardly observed in WT. In this system, we observed that endogenous DSCAM-HA colocalized with PSD95 (excitatory postsynapse marker) and Gephyrin (inhibitory postsynapse marker), suggesting that endogenous DSCAM is localized at postsynapses of both excitatory and inhibitory

synapses. This has been presented in Fig. 1g, h, of the revised manuscript. This is the first case of detecting endogenous DSCAM at mammalian brain synapses by immunostaining, although a similar localization pattern was observed by overexpressing His-tagged DSCAM in cortical neurons [Chen *et al.*, *J Neurosci*, 2022, 42(4)].

Furthermore, we are trying to generate another mouse KI line, in which a more sensitive probe, 'spaghetti-monster' fluorescence protein (smFP) [Viswanathan *et al.*, *Nat Methods*, 2015, 12(6)] is tagged with DSCAM protein. However, it takes a long time to generate this line, and we could not include the result using this line in the revised manuscript. Because of the larger molecular size of this smFP (about 40 kDa) compared to the tandem PA tag and HA tag, we cannot guarantee that it will not affect the molecular localization of DSCAM but we are currently pursuing this experiment because we have no other options. Therefore, in our revised manuscript, we decided to show the localization of endogenous DSCAM-HA in primary cultured hippocampal neurons. We also showed the concentration of DSCAM in the synaptosomal fraction from the WT cerebellum (Fig. 1f), suggesting DSCAM localization at cerebellar synapses. If the reviewer thinks that the data using smFP-tag KI mice is essential, we can include them in the future, although it may take more than half a year. We described it in the Results section of the revised manuscript.

Pages 7-8, lines 158–169, in Result section

“Next, we have created a knock-in mouse line designed to express DSCAM protein tagged with tandem HA (Fig. 1g,h, Supplementary Fig. 2a-f). However, the signals with anti-HA antibodies were very weak in the postnatal cerebellar slices of the KI mice (*Dscam*^{HA/HA}). Thus, we analyzed the localization of endogenous DSCAM-HA protein in primary cultured neurons from *Dscam*^{HA/HA} mice. Unfortunately, in the cultured primary neurons from the cerebellum, we detected the background signals with the anti-HA-tag antibody in WT and *Dscam*^{HA/HA} mice (Supplementary Fig. 2f). However, in primary cultured neurons from the hippocampus, we detected significant signals in *Dscam*^{HA/HA} mice, which were hardly observed in WT (Fig. 1g,h). We observed that endogenous DSCAM-HA colocalized with PSD95 (excitatory postsynapse marker) and Gephyrin (inhibitory postsynapse marker) (Fig. 1g,h), suggesting that endogenous DSCAM is localized at postsynapses of both excitatory and inhibitory synapses.”

Pages 22, lines 533–569, in Methods section

“Rigorous selection/optimization of the tag-insertion site in DSCAM

The selection and optimization of the HA-tag-insertion site of DSCAM was described in our previous paper⁴⁸ Because DSCAM has a signal peptide for membrane insertion at the N-terminus and a PDZ binding motif at the C-terminus, terminal fusion of artificial tag may inhibit the DSCAM functions. To determine the tag-insertion site, amino acid sequence for mouse DSCAM protein was compared with DSCAM from other species using the web-based protein structure prediction tool Jpred 4 (<http://www.compbio.dundee.ac.uk/jpred/>) to find regions with low stringencies. The selected region was analyzed using the web-based CRISPR design tool CRISPOR (<http://crispor.org>) in order to choose the guide RNA sequence with minimized off-target effects. Tandem HA-tag was inserted into the cytoplasmic region to obtain the *Dscam*^{HA} KI mouse line, which was confirmed to show no changes in total DSCAM expression levels. Mice homozygous for *Dscam*^{HA} were viable and fertile, exhibiting normal development with regard to body size and weight, and did not present any abnormalities throughout the study period. This carefully evaluated line was used accordingly in this study.

Generation of Dscam^{HA} KI allele by CRISPR/Cas9-mediated genome editing

Two parts of CRISPR guide RNA, *Dscam*-crRNA (5'-TTGTTAAACCGGGGCGCACCGTTTTAGAGCTATGCTGTTTTG-3') and tracrRNA (5'-AAACAGCAUAGCAAGUUAUUUUUUAAGGCUAGUCCGUUAUCAACUUGAAAAAGUGGCACCGAGUCGGUGCU-3') were chemically synthesized and purified by reversed-phase column purification (FASMAC), as described in our previous paper⁴⁸. Recombinant Cas9 protein (EnGen Cas9 NLS) was purchased from New England Biolabs (NEB; Ipswich, MA). Cleavage activities of the guide RNAs with Cas9 protein were evaluated in our previous paper⁴⁸. The single-strand DNA (ssDNA) donor containing tandem HA-tag sequence flanked by 52 bp homology arms on both sides was designed as follows. The left (5'-CATACACCTGCCTCCATACCTACGAATGGACTTCTTGTTAAACCGGGGCGCA -3') and right (5'-TCTGGGGTTCCAAGCAGCTTGTCTAAACTCAGGTCCCTGCTGGTGCTGG -3') homology arms were connected to tandem HA-tag sequences (5'-TATCCTTATGACGTACCTGACTATGCATACCCATACGATGTTCCAGATTACGCT -3'). The ssDNA was chemically synthesized by Eurofins Genomics (Luxembourg). The guide RNAs, Cas9 proteins, and donor ssDNA were injected into

the oviductal lumen of ICR mouse and electroporated into zygotes by improved genome-editing via oviductal nucleic acids delivery (*i*-GONAD) method⁹². Using PCR with three sets of primers and sequence analyses, 5 out of 9 pups were confirmed to carry the correct HA-tag KI allele. A pair of primers, F (5'-TTCATGTCTTGGGTGGGCTC-3') and R (5'-TTTGCGCTGTCTGTGGTTTC-3'), were used to genotype the progenies, as described in our previous paper⁴⁸."

2) a better temporal resolution: How do the phenotypes arise during development.

Response: In response to the reviewer's suggestion, we included three types of experimental data in the revised manuscript and described them in the Result section as follows:

(i) In the original manuscript, we reported that the formation of CF synapses in the *Dscam*^{del17/del17} cerebellum decreased from P15 to P30. In the revised manuscript, we newly examined the projection of CF synapses at P22 and eventually found no change in the number of CF synapses at P22 between WT and *Dscam*^{del17/del17}. These data were summarized graphically and included in Supplementary Fig. 3d,e of the revised manuscript.

(ii) We have examined in detail the changes observed in cerebellar morphology at P7, P15, and P30, which is presented in Supplementary Fig. 3b. We found that distal lobuli were unusually longer than those of WT cerebella at any developmental stages tested (P7, P15, and P30). We described it in the Results section of the revised manuscript.

Pages 8, lines 174–179

"The distal portion of the caudal lobuli is unusually longer⁴⁹ compared to those of the WT from P7 to P30 (Supplementary Fig. 3b). The dorsal midbrain of *Dscam*^{del17/del17} mice was reported to be hypertrophic^{48, 50}. Therefore, the enlarged midbrain may compress the cerebellum in the caudal direction, resulting in stretching of the lobuli at the caudal side, although other possibilities remain."

(iii) We examined in detail the changes seen in cellular and synaptic developmental processes between P7 and P30. These data are presented in Supplementary Fig. 4, 5, and Supplementary Table 1 in the revised manuscript. We described it in the Results section of the revised manuscript.

Page 9, lines 197–211

“At several developmental stages (P7, P15, and P30), we performed immunostaining for various markers of specific cell types, synapses, or apoptosis (Supplementary Fig. 4, 5, Supplementary Table 1). During development (from P7 to P30), prominent changes were hardly observed in the morphology of Purkinje cells (Calbindin, a marker for Purkinje cells; Fig. 2a, Supplementary Fig. 4a,c, 5g), the density or distribution of vGluT1 (a marker for PF terminals; Supplementary Fig. 4a,b) or vesicular GABA transporter (vGAT, a marker for inhibitory fiber terminals; Supplementary Fig. 4c,d, 5g), or in the morphology of Bergmann glial cells (GFAP, a marker for Bergmann glia; Supplementary Fig. 4e, 5h). The number of Caspase-3-positive apoptotic cells (Supplementary Fig. 5a) and Ki67-positive proliferative cells (Supplementary Fig. 5b) remained unchanged in *Dscam*^{del17/del17} cerebella at all the tested developmental stages. The number of Olig2-positive cells, including oligodendrocytes and OPCs, decreased significantly in *Dscam*^{del17/del17} cerebella at P7 but not at P15 and P30 (Supplementary Fig. 5c, Supplementary Table 1). DSCAM may have some roles in the proliferation of OPCs, as suggested by the relatively high *Dscam* transcript levels in OPCs (Fig. 1a, Supplementary Fig. 1a).”

1-2

Page 7 line 147. It is essential to provide detailed information how the fusion-protein was generated and whether potential impairment of function/localization can be excluded.

Response: We agree with the reviewer’s opinion as to the information on the fusion protein. The creation method for the fusion protein was described in the Methods section of our previous paper [Arimura *et al.*, *Sci Adv*, 2020, 6(36)] as shown below. “To generate Dscam-mEGFP, the PCR product of monomer EGFP (mEGFP) with ten glycine repeat linkers (Gly) was inserted in-frame into the Dscam full-length coding sequence at the MroI restriction enzyme site (in Fibronectin type III domain [FnIII]), which is a usable and unique enzyme site within Dscam, using the In-Fusion HD Cloning Kit (Fig. 1f).”

The localization and function of the fusion protein were assessed as below in the Results section of our previous paper [Arimura *et al.*, *Sci Adv*, 2020, 6(36), in Extended Data Fig. 5A-D] as below.

“We validated DSCAM-mEGFP in terms of its dimerization potential with non-tagged DSCAM, its binding ability to RapGEF2 (a DSCAM interacting protein, see below), and its subcellular localization upon transfection into cultured cells, all of which were the same for untagged DSCAM (Supplementary Fig. 5a,b,d)”

For better understanding, we added the following sentences to the Methods and Results in the revised manuscript.

Page 7, lines 146–148, in Result section

“We previously reported that subcellular localization, dimerizing potential, or protein binding ability was not different between non-tagged DSCAM and DSCAM-mEGFP⁴⁸.”

Page 21, lines 514–515, in Methods section

“The procedure to generate DSCAM-mEGFP expression plasmid has been described previously⁴⁸.”

1-3

For DSCAM knock-out experiments it is important to describe and summarize what phenotypic traits have actually been examined (examining cerebellar development). Given the known functions, it would seem important to determine potential self-avoidance defects that may indirectly impact on synaptogenesis.

Response: According to the reviewer’s comment, we have summarized the data analyzed so far in a table and added it to the revised manuscript as Supplementary Table 1.

It is well known that DSCAM is involved in homo-repulsion or self-avoidance in various developmental cellular events. Since *Dscam* gene expression is prominent in Purkinje cells and CF neurons in the ION (Fig. 1a,b), there are two possibilities wherein self-avoidance may occur. (i) At the CF synapse between the Purkinje cell dendritic spines and the climbing fiber terminal of the ION neurons, where DSCAM-DSCAM interaction can occur, and thus self-avoidance is assumed to occur. However, if self-avoidance between pre- and post-synapses is important, one would expect the loss of *Dscam*, resulting in an increased formation of CF synapses but this did not occur; rather, it decreased. (ii) The second possibility is the involvement of self-avoidance in Purkinje cell sister dendrites. If self-avoidance between sister dendrites is important, one would

expect the loss of Dscam to increase the entanglement of dendrites. However, no significant morphological abnormalities were observed in the dendrites of Purkinje cells in *Dscam*^{del17/del17} mice (Supplementary Fig. 4a,c).

Since *Dscam* expression was hardly observed in granule cells and Bergmann glia (Supplementary Fig. 1a), DSCAM-DSCAM interactions in PF synapses and/or synapse-glia are unlikely. Based on the above considerations, self-avoidance does not seem very likely to be involved in the formation of CF synapses, although we cannot completely rule out other possibilities.

Page 7 line 163: seems to be the wrong citation, as this paper (Fuerst *et al.*) examines the role of DSCAM only in the visual system.

Response: In Supplementary Fig. 2e,f of the cited paper (Fuerst *et al.*, *Nature* 2008, 451), the morphology of the cerebellum for control and *Dscam* mutant (*Dscam*^{del17/del17}) was presented and discussed. Therefore, we have cited this reference.

Page 7 line 167: phenotype needs to be described precisely. What is the cellular phenotype? Are there less cells, more apoptosis, etc. Quantification would be very helpful.

Response: Considering the reviewer's suggestion, we added immunostaining data (Supplementary Fig. 5) and corresponding statistical data (Supplementary Table 1) to the revised manuscript. We performed immunostaining with Parvalbumin (Interneuron marker), Iba1 (Microglia marker), Olig2 (oligodendrocytes and OPC marker), Pax6 (granule cell marker), and Calbindin (Purkinje cell marker) to measure the cell density of distinct cell types. We performed immunostaining for Cleaved Caspase3 and KI67 to estimate apoptosis and proliferation of cells, respectively. We found no significant differences in interneurons, densities of microglia, granule cells, Purkinje cells, or apoptotic and proliferating cells in *Dscam*^{del17/del17} mice relative to the WT mice. However, a slight decrease in OPCs was found in P7 *Dscam*^{del17/del17} cerebella. Statistical data are summarized in Supplementary Table 1.

Page 4, line 72/73: poor sentence - the term localization rather than recognition should be used

Response: According to the reviewer's suggestion, we have replaced the word "recognition" with "localize" in the revised manuscript.

Page 4, lines 66–67

"However, how these functionally critical transporters localize the synapses and function near the peri-synaptic area remains largely unknown."

Page 14 line 343: "intracellular heterophilic association". This is incorrect, presumably, the authors wanted to use the term "intercellular". That section needs to be clarified as it actually refers to a proposed mechanistic model (not yet supported by data).

Response: Thank you very much for pointing out our mistake. We have corrected the sentence in the revised manuscript.

Page 15-16, lines 368–370

"We found that the intercellular heterophilic association of DSCAM with GLAST between postsynapse and enwrapped astrocytic Bergmann glia was required for proper CF synapse formation."

Throughout the entire text, I do not understand what the authors refer to when they use the term "relocate". Are they referring to dendritic targeting?

Response: We looked in the original manuscript but could not find the word "relocate" or any related words such as "relocation." Instead, we found the word "delocalization." To clarify the meaning, the phrase "away from the synaptic cleft" has been added at the end of the sentence in the revised manuscript where the word "delocalization" was used.

Page 12, lines 285–286

"These results suggested that the depletion of Dscam caused the delocalization of GLAST away from the synaptic cleft."

Page 5, Line 91-94. Sentence makes no sense. What is "unbalanced localization"?

Response: We are sorry that we did not explain the above points well enough in the previous manuscript. We have modified the sentences accordingly in the revised manuscript

Page 5, lines 85–90

“Interestingly, GLAST is localized at the cell membrane of the Bergmann glia in the cerebellum but its distribution is not uniform and is uneven. GLAST shows strong localization to areas in contact with neuropils and synapses but localizes weakly to other areas, such as the pia mater facing regions and dendrite shafts of the Purkinje cells¹⁵; however, the molecular mechanism underlying the uneven localization of GLAST remains unclear.”

Reviewer #2 (Remarks to the Author):

Although the authors tend to focus on the climbing fiber-Purkinje cell synapse (CF), mostly in terms of its formation, the fact that the absence of DSCAM impairs the parallel fibers-Purkinje cell synapse (PF) might prove to be an important finding taking into consideration that it is thus synapse (PF) the most abundant glutamatergic synapse in the whole brain. In this context, it might be important to discuss in a more deep detail, the fact that DSCAM is required for motor learning but not for motor coordination.

Response: Thank you for your valuable comment. Accordingly, we added a paragraph in the Discussion on motor learning and motor coordination.

Page 18-19, lines 444–456

“In *Pcp2^{Cre}-cKO*, we found the defect in motor learning (Fig. 7a-d) but in motor coordination such as during walking, grasping food, and running in daily life. Similar results have been reported previously. mGluR1b-rescue mice, wherein the splice variant of mGluR1 is expressed in mGluR1-null, exhibited impairments in CF synapse elimination, induction of long-term depression (LTD), and delay in eyeblink conditioning, another form of cerebellum-dependent motor learning but showed normal motor coordination⁷⁸. Similarly, a single recombinant CBLN1 injection transiently restored PF- LTD and eyeblink conditioning in adult *Cbln1*-null mice without affecting the CF innervation pattern⁷⁹. In *C1ql1*-null and *PC-Bai3*-null mice, motor learning and PF-LTD were severely impaired but motor coordination was normal⁹. Taken together, PF-LTD and motor learning, but not motor coordination, seem to be closely correlated⁸⁰. Therefore, future studies should assess whether and how DSCAM proteins mediate synaptic plasticity in PF, CF, and other synapses essential for memory and learning.”

2. Taking into consideration that Bergmann glia express very low levels of the other glial glutamate transporter Glt-1, it is surprising that disrupted GLAST localization in the DSCAM-depleted animals the most relevant finding is the failure to establish proper CF synapses. One would expect a severe neurodegenerative scenario due to the excessive glutamate present into only in the synaptic cleft but also the spilled over neurotransmitter. The authors are prompted to provide an explanation in the Discussion.

Response: As the reviewer pointed out, it is widely known that excess glutamate induces neurodegeneration, and we thought that we should mention this possibility in our mice. Since we partially discussed spillover in the original manuscript, we have added the following sentences highlighted in yellow to the Discussion section.

Page 17, lines 409-430

“Differential effects on PF or CF synapses due to the depletion of Dscam.”

In this study, the application of CTZ significantly increased the amplitude ratio of PF-EPSCs in *Dscam*^{del17/del17} mice, whereas there were no significant differences in the amplitude and decay time of PF- and CF-EPSCs, as well as the amplitude ratio of CF-EPSCs between WT and *Dscam*^{del17/del17} mice under both control and CTZ conditions (Fig. 3e–j). In contrast, in GLAST-KO mice, CTZ treatment increased the peak amplitude of the PF- and CF-EPSCs and prolonged its decay beyond the value observed in WT mice^{14, 75}. The reason for this phenotypic discrepancy between *Dscam*^{del17/del17} and GLAST-KO mice is probably that the control of GLAST by DSCAM is restricted to the peri-synaptic area. Usually, GLAST is localized not only in the inter-synaptic cleft but also in the gap between the pre- or post-synapse and Bergmann glia, where it is thought to prevent spillover of superfluous glutamate to neighboring synapses⁷⁵. **Impaired glutamate uptake has been implicated in neurotoxicity and neurological diseases^{76, 77}. In *Dscam*^{del17/del17} mice, however, we observed no significant cerebellar atrophy due to cell death and no significant change in the number of Caspase3-positive cells, a marker of apoptosis (Supplementary Fig. 5a, Supplementary Table 1), and the amount of membranous GLAST was comparable to that of WT (Fig. 3m, Supplementary Fig. 7a–d). Therefore, the complete loss of GLAST may markedly increase superfluous free glutamate and EPSC amplitudes, whereas partial reduction of GLAST near the peri-synaptic area by loss of DSCAM may not incur a dramatic effect on EPSCs. The decrease in CF synapse seen in both GLAST KO and *Dscam*^{del17/del17} mice may be due to the heterosynaptic competition between PF and CF synapses, as described above, and not due to toxicity from excess free glutamate.”**

3. As the authors correctly point out, DSCAM binds to Glt-1. Being Glt-1 roughly 1% of total brain protein the authors might consider to speculate if DSCAM-Glt-1 is also responsible of a proper Glt-1 localization.

Response: We also acknowledge that it is an important point. However, we intend to discuss it in the next paper. Therefore, we mentioned it briefly in the Discussion as follows:

Pages 20, lines 483–493

“DSCAM-GLAST interaction and neuropsychiatric disorders

To date, the DSCAM molecule has been known implicated in neuropsychiatric disorders such as schizophrenia and autism^{32, 33, 34, 35, 36, 37}. Many molecules involved in such disorders have been reported to be involved in synaptogenesis and transmission^{33, 34, 35, 36, 37}. Among them, impaired glutamate removal is known to cause abnormal synapse formation, neuronal cell death due to excessive input, and behavioral abnormalities in mice⁹⁰. Genetic mutations in glutamate transporters have also been reported in human genome wide sequencing of psychiatric disorders^{33, 34}. These results suggest that the interaction between DSCAM and GLAST is involved in neuropsychiatric disorders. Whether the localization of GLT-1, another glutamate transporter, is also regulated by DSCAM is a subject for future study. In the future, we should focus on how these DSCAM functions are regulated throughout the brain regions.”

4. One of the authors main conclusions is that augmented free glutamate is responsible of the impairment of CF synaptogenesis, it is important at the authors elaborate in a plausible mechanism (Page 15, second paragraph).

Response: Considering the reviewer’s suggestion, we added our hypothesis to the Discussion.

Pages 16–17, lines 392–407

“The impairment of CF synapse translocation in the *Dscam*^{del17/del17} cerebellum was prominent from P15. PF synapse activity is required for heterosynaptic competition between PF and CF synapses from P15 onwards⁴. The territory overlaps and the subsequent segregation of PF and CF synapses on Purkinje cell proximal dendrites are driven by the elimination of PF synapses from the overlapping portions as well as dendritic translocation of the single strengthening CF from P15 to P30. In *Dscam*^{del17/del17} mice, CF translocation weakened significantly along with the unusual formation of PF synapse-like cobbled dendritic spines at the proximal dendrites of Purkinje cells (Fig. 2a–c). Territory transition is arrested in mutant mice lacking mGluR1–Gaq–PLC4β–PKCγ signal transduction, which is important for input from PF

synapses⁴. The GluD2–CBLN1–NEUREXIN system is also critical for the formation and maintenance of PF synapses, and thus, contributes to the establishment of properly segregated CF territories^{73,74}. Although we could not detect the unusual change in mGluR1 for PKC γ signaling or the changes in CF translocation-related transcript expression in Purkinje cells in *Dscam*^{del17/del17} cerebella (Supplementary Fig. 8a–j, 9a–d), these reports implied that abnormal PF synapse input and/or formation caused impairment of CF synapse translocation and formation in *Dscam*^{del17/del17} mice. Therefore, one possible hypothesis is as follows: impaired binding of DSCAM-GLAST at PF synapses promotes an increase in free glutamate at synaptic cleft and gives excessive input to the post-synapse of Purkinje cells. This excessive input may somehow inhibit pruning on primary dendrites at PF synapses and cause abnormal stabilization of PF synapses, thereby preventing dendrite CF translocation and enhancing CF formation from P15 to P30.”

5. As depicted in Extended Data Fig 10 panel C, DSCAM is needed for the proper GLAST localization and function. This is not a minor issue and the scenario es clearly more complex than the one depicted in the referred figure. In fact, this is a major finding, please discuss this in the context of cerebellar function, since failure of a proper glutamate uptake triggers a severe energetic deficit (see *Neuropharmacology* 192 (2021) 108602 for a review., DOI: 10.3389/fphys.2021.825816).

Response: We have addressed your points in the Discussion as follows:

Page 20, lines 475–481

“Astrocytes also contribute to memory, neuroprotection, and homeostasis via energy metabolism in the brain⁸⁸. Glutamate taken up by astrocytes is used in the TCA circuit⁸⁹, and some of it is converted to α -ketoglutarate by glutamate dehydrogenase and utilized for ATP production in the mitochondria. This suggests that the delocalization of GLAST may cause an energy deficit associated with impaired glutamate retrieval in the cerebellum. Therefore, further studies may be required to investigate phenotypes caused by abnormal energy metabolism in the *Dscam* cKO cerebella.”

Reviewer #3 (Remarks to the Author):

In Fig 4 authors investigate whether GLAST interacts with DSCAM performing CO-IP of endogenous proteins in cerebellar synaptosomes and overexpressed proteins in COS-7 cells. This is an important figure as it demonstrates novel DSCAM interaction with astrocytic protein. Co-IPs are usually done both ways to rigorously show that proteins can precipitate each other. Could authors please show that DSCAM can IP GLAST?

Response: We have already confirmed endogenous GLAST and DSCAM coprecipitation using the synaptosomal fraction in both ways (Fig. 4a in the original manuscript and the revised manuscript). This shows that DSCAM can IP GLAST *in vivo*.

Perhaps because DSCAM is a large molecule with a weight exceeding 250 kDa, its expression was low when transfected into COS cells. Furthermore, we confirmed that DSCAM expression was even lower when cotransfected with GLAST. Therefore, it was difficult to IP DSCAM efficiently in cultured cells. Because we were able to confirm the binding of DSCAM to GLAST in our IP experiments on endogenous DSCAM in two ways, we did not present IP experiments for DSCAM in cultured cells in this paper. If the reviewer thinks this experiment is essential, we would consider using various types of cultured cells and would require some time to study the conditions.

Authors also state in Fig legends that this was repeated 3 times (n=3). IP in Fig 4a indicates a very weak interaction (DSCAM band is large in the input and very faint in GLAST-IP) -therefore showing all three independent repeats would be important to support the conclusion.

Response: According to the reviewer's suggestion, we have added the remaining two experimental data to Supplementary Fig. 11 in the revised manuscript.

In Fig 4c IP is done with overexpressed proteins in COS7 cells- could authors please indicate which of many bands is specific for the GLAST?

Response: We are sorry for the confusion caused due to the picture in the original manuscript. In the right-upper panel of Fig. 4c, non-specific bands were observed in the low molecular weight region. Therefore, we have added arrowhead to Fig. 4c of the

revised manuscript to indicate the specific band. The other bands correspond to DSCAM proteins produced from the expression construct in Fig. 4b. This has been added to the figure legend.

2. In the abstract authors claim that “Dscam-mutant mice mimicked loss-of-function mutations in the astrocytic glutamate transporter GLAST expressed in Bergmann glia.” This is not accurate -GLAST loss of function mice have different phenotype from DSCAMdel7/del7 mice described in this manuscript (i.e. impaired glial wrapping of synapses vs no changes here, multiple CF innervation vs no changes here, altered CF-EPSC vs no changes here).

Response: As you point out, *Dscam*^{del17} mice do not show all of the phenotypes that are seen in GLAST KO mice. Therefore, we have changed the sentence in the Abstract.

Page 3, lines 49–51

“*Dscam*-mutant mice showed defects in CF synapse translocation as is observed in the case of loss-of-function mutations in the astrocytic glutamate transporter GLAST expressed in Bergmann glia.”

3. Authors next state “. These mice showed impaired glutamate clearance by the delocalization of GLAST, which complexed with the extracellular domain of DSCAM.” This is not fully supported by results. First, to support impaired glutamate clearance authors should manipulate GLAST (i.e. use blockers of its function such as TBOA and enhancers such as riluzole) and show differential effect on wt and DSCAMdel7/del7 PF –PC synapses.

Response: According to the reviewer’s suggestion, we investigated whether riluzole administration to *Dscam*^{del17/del17} cerebellar slices rescue the amplitude ratio (+CTZ/Control) of PF-EPSCs in whole-cell patch-clamp recordings. As we showed in the original manuscript, in the presence of 100 μM CTZ, a significant difference was detected in the amplitude ratio of PF-EPSCs (Fig. 3j) between WT and *Dscam*^{del17/del17} mice. However, in the presence of 100 μM CTZ with 60 μM riluzole (for 10 min), we found that there were no significant differences in the amplitude ratio (+CTZ+riluzole/Control) of PF-EPSCs (see below, figure for reviewer 1). These results suggest that riluzole treatment counteracted the difference in free glutamate

concentration at the synaptic cleft between WT and *Dscam*^{del17/del17} mice. As we mentioned in the original manuscript, and as you mentioned below (comment #4), riluzole has many different effects, and this result may not only be caused by the enhancement of GLAST. However, our present results also revealed that: (1) *Dscam*^{del17/del17} mice showed the change of GLAST localization away from synaptic cleft, and (2) DSCAM protein associates with GLAST at the extracellular domain. Furthermore, the defects of CF synapse translocation on the proximal dendrite of Purkinje cell in *Dscam*^{del17/del17} cerebellum were also observed in *Glast* KO mice (Miyazaki T *et al.*, *Eur J Neurosci*, 1998, 10). Therefore, we thought that impaired glutamate clearance occurred in *Dscam*^{del17/del17} cerebellum, and that riluzole affects the glutamate clearance into Purkinje cell-specific *Dscam* cKO mice.

Figure for reviewer 1:

As for TBOA application to rescue the effect of riluzole, we are concerned about the non-specific, inhibitory effects of TBOA on the EAATs (Kanai *et al.*, *Mol Aspects Med*, 2013, 34). Since EAAT4 is expressed in Purkinje cell dendritic spines, TBOA treatment in the cerebellum slice would flood glutamate at the synaptic cleft through the inhibition of glutamate clearance by EAAT4 at the post-synapse in Purkinje cells, as well as by GLAST in the Bergmann glia. Riluzole could affect GLAST, however, this has not been confirmed whether riluzole could affect EAAT4 activity (Fumagalli *et al.*, 2008, *Eur J Pharmacol*, 578). Therefore, if TBOA application cannot rescue the effects of riluzole on *Dscam*^{del17/del17} PF ratio, we cannot determine whether this failure to rescue was due to the inhibition of EAAT4 which may not be activated by riluzole, or due to the failure to reduce free glutamate in the synaptic cleft of *Dscam*^{del17/del17} mice neurons through the activation of GLAST by riluzole. Therefore, we have not done this experiment.

To avoid misunderstandings, we changed the description in the Abstract.

Page 3, line 51-53

“These mice suggested impaired glutamate clearance and the delocalization of GLAST away from the synaptic cleft. GLAST complexes with the extracellular domain of DSCAM.”

Second, to support delocalization of GLAST it is critical to show that wrapping of BG processed around synapses is not altered and that GLAST expression is not altered.

Response: In electron microscopy analysis, we observed that *Dscam* mutation did not affect Bergman glia synapse wrapping (Supplementary Fig. 10e in the revised manuscript, Extended Data Fig. 8e in the original manuscript), the length of PSD (Supplementary Fig. 10f in the revised manuscript, Extended Data Fig. 8f in the original manuscript), and the number of Bergman glia process edges close to PSD (Supplementary Fig. 10g in the revised manuscript, Extended Data Fig. 8g in the original manuscript). We show that the wrapping of BG processed around synapses is not altered in Supplementary Fig. 10e and g in the revised manuscript and in Extended Data Figs. 8e-f of the original manuscript.

In Fig 3L authors show cumulative frequency of GLAST signal from the PSD edge. This result is confusing as 100 was reached at 0.4 μ m distance from the nearest PSD edge. Yet based on extended data 8g only approximately 20% of synapses had BG processes that are closer than 0.4 μ m to PSD. Based on that result only 20% synapses had a chance to have GLAST within 0.4 μ m from PSD (and even that if and only if GLAST was localized at the PSD-closest part of BG processes). Could authors please explain this discrepancy? Could authors also show the average distance of GLAST to PSD per mouse?

Response: We apologize for any confusion due to our inappropriate description in the original manuscript. In Fig. 3l, we used the two groups classified in Supplementary Fig. 10g in the revised manuscript (Extended Fig. 8g in the original manuscript). One is the group of glial fiber is closer to the PSD edge by 0.4 μ m on both synaptic sides (indicated “2” as dark gray in legend, about 25%) and another is on one synaptic side (indicated “1” as light gray in legend, about 45%). Thus, we used 70% of the synapses that we examined. The remaining 30% were excluded because the distance between the glial fiber and the PSD edge was more than 0.4 μ m to begin with, which was

considered distant regardless of the delocalization of GLAST. To clearly describe this, we changed the legend of Supplementary Fig. 10g in the revised manuscript.

Could authors also show the average distance of GLAST to PSD per mouse?

Response: Following the reviewer's suggestion, we added the average distance of GLAST to PSD edge per mouse in Fig. 3l.

In extended data Fig 5 authors investigate whether GLAST expression is altered in DSCAMdel7/del7 mice. For WB in extended data fig5b please show all three WT and DSCAMdel7/del7 samples, include loading control (actin, tubulin etc.) and quantify results.

Response: According to the reviewer's suggestion, we have added WB data to the original data (n=2); finally, n=3 for immunoblot data in Supplementary Fig. 7b, c in the revised manuscript. We have also added loading control data to Supplementary Fig. 7b, c. Quantitate analyses in c were normalized against the protein content of β -actin.

In extended data Fig 5c and d –quantification indicates decrease in GLAST puncta but it is not stat significant. However large variability in WT data would indicate that larger number of samples than used here is needed to reach statistical significance. Authors should use power formula and SD of their WT results to calculate number of samples needed.

Response: We used the power formula to determine if the number of samples analyzed in Extended Data Fig. 5d in the original manuscript was adequate. Then, we realized that the power formula was relatively low (power=26.9). We thought that this low-power formula might be due to the high variability in the WT data and considered increasing the sample size for WT. The original manuscript data was imaged with a Zeiss LSM 780 confocal microscope system in 2018 but this microscope has already been replaced by a new one (Olympus). Therefore, it turned out that newly acquired data could not be merged with the old data because the imaging conditions were different. Because most of the *Dscam*^{del17/del17} mice died within 24 h of their birth (most pups did not have milk in their stomachs) (Amano *et al.*, *J Neurosci*, 2009, 29), it is difficult to collect many of these mice for this and other experiments suggested by the three referees. However, we attempted the analysis again with newly prepared

samples (WT: n=8, *Dscam*^{del17/del17}: n=4) and a new microscope system (Olympus). The power formula of the experiment was 32.91 %. This low estimation seems to be caused by the properties of GLAST localization as reported previously (Ageta-Ishihara *et al.*, *Nat Commun*, 2015, 6; Miyazaki *et al.*, *Proc Natl Acad Sci U S A*, 2017, 114) (see below: Figure for reviewer 2): some areas show punctate labeling, but other areas lack it because of the thick primary dendrite or involvement of the thinner filamentous structures aligned to the Purkinje cell dendrite. Therefore, we retracted the Extended Data Fig. 5d in the original manuscript and added a newly examined graph indicating the contents of GLAST proteins estimated by western blot analyses (Supplementary Fig. 7c).

Figure for reviewer 2:

4. Abstract also states that “Riluzole-induced reduction of free glutamate at the synaptic cleft rescued the impairment of CF synapse formation in Purkinje cell-selective *Dscam* deficient mice”. This is also not supported by data.

First riluzole has many different effects and authors have not examined whether riluzole indeed reduces free glutamate at the synaptic cleft. Experiment proposed above (add riluzole/TBOA to ephys to rescue PF ratio) could address whether riluzole affects glutamate clearance in *DSCAM* del7/del7 mice. Second, as riluzole did not rescue distal CF innervation this should be stated in the abstract, or clarified that riluzole rescued only proximal impairment of CF synapse formation.

Response: We agree with the reviewer’s comment that Riluzole has a lot of effects on synapses through distinct target molecules. Thus, following your suggestion, we changed the description in the Abstract.

Page 3, lines 53–55

“Riluzole, as an activator of GLAST-mediated uptake, rescued the proximal impairment in CF synapse formation in Purkinje cell-selective *Dscam*-deficient mice.”

Response: As for the TBOA application, please see our comment above.

5. . Following information should be added:

1. Methods states “No blinding was performed”. Please clarify why blinding was not used in this study.

Response: We are sorry that the description in the original paper was inappropriate. In most of our experiments, especially in the electrophysiological and mouse behavioral experiments, it was difficult to perform them in a blinded manner, because a single researcher with advanced skills had to concentrate on a single experiment. For general immunostaining and western blotting, a single researcher was in charge throughout a single experiment but the data from the second and third experiments were analyzed by a different researcher to confirm the reproducibility of the results. Furthermore, in most cases, we measured the data automatically by using some detectors or microscopy. We believe such experimental strategies would significantly improve the elimination of preconceived notions. In this revised manuscript, a table summarizing whether the experiment was performed in a blind, automatic measurement, or non-blind manner (shown below) is provided; please see Supplementary Table 2 in the supplementary information.

2. In Methods please add description of VGLU2 puncta quantification.

Response: We added the description of quantification on cell count, dendritic height, and the puncta number of vGluT2 as follows:

Pages 26, lines 643–650

“The number of cells, dendritic height, and the height and number of vGluT2 puncta were measured using the Fiji/ImageJ 2.3.0/1.53. For measurements of the height and puncta number of vGluT2, puncta with sizes greater than $0.5 \mu\text{m}^2$ and smaller than $13 \mu\text{m}^2$ were defined as vGluT2-positive CF presynapses whereas puncta smaller than 0.5

μm^2 or located outside the dendritic shafts. The configured puncta size was automatically counted using “Analyze Particles.” The density of vGluT2 puncta was automatically measured and calculated using “Trainable Weka segmentation”, “Auto Threshold”, and “Analyze Particles” modules in Fiji/ImageJ with slight modifications.”

3. In Figures 2, 3, extended data 4, 7, and 9 please include number of mice used (if possible in the brackets by the mouse genotype as done in the rest of the figures?)

Response: The number of mice is listed in the brackets or on the column by the mouse genotype in each figure according to the comments.

4. In methods authors states that for CF-EPSC recordings AMPA R antagonist was used. Could authors clarify how can then cyclothiazide (CTZ) which reduces desensitization of AMPA R work in such condition?

Response: We apologize for the lack of an explanation. It is well known that usual EPSCs in Purkinje cells have very large currents and are hard to record repeatedly due to difficulty in returning to baseline. In this study, the low concentration (500 nM) of NBQX as an AMPA R antagonist was included in the external solution to block some AMPA R and reduce EPSC size (Liu and Friel, *J Physiol*, 2008, 586). The higher concentration (50 μM) of NBQX is required for the complete blockade of AMPA R (Kyung-Seok *et al.*, *Nat Neurosci*, 2020, 23), so that the lower concentration of NBQX does not disturb the examination of CTZ’s effects on EPSC in *Dscam* mutant Purkinje cells. We add a reference (Liu and Friel, *J Physiol*, 2008, 586) to the Method section.

5. In Discussion there is paragraph titled “CF synapse selective effect by the depletion of *Dscam*” in which authors discuss that PF-EPSC is affected and not CF-EPSC and speculate on why. Perhaps title should be changed to indicate this.

Response: Page 17, line 409

Following your comments, we changed the subtitle of this paragraph to “Differential effects on PF or CF synapse due to the depletion of *Dscam*.”

6. In Methods for synaptosome preparation authors write “The samples were centrifuged at 1,200 X g for 10 min. The collected supernatant was centrifuged at 1,500

X g for 20 min, and the resultant pellets were used for following assays.” Is the second rotation speed (1500) a typo (i.e. should it not be larger?)

Response: As you pointed out, this is a typo. In fact, it was centrifuged at 15,000 g. We have corrected the description in Methods (Page 27, line 665).

REVIEWER COMMENTS

Reviewer #2 (Remarks to the Author):

The authors have successfully addressed all the points raised by this reviewer. The revised version is now suitable for publication in Nature Communications.

Reviewer #3 (Remarks to the Author):

1. Reviewer appreciates including additional two co-IP experiments in supp fig 11. Unfortunately, even with both Fig 4A and supp fig 11, it is difficult to evaluate these results as DSCAM bands are too faint. Input levels for both DSCAM and Glast are also missing. Since this is an important data reviewer suggests in addition to adding input, to perform co-IP with the known DSCAM interactors as control as well as using synaptosomes from DSCAM Pcp2 KO mice Fig 4A and supp fig 11 please include input.
2. Because authors suggest the differential effect on DSCAM deletion on PF and CF synapses-for the fig 3 m they need to separately investigate delocalization of glast from synapses in lower (I) and higher (III) where most synapse are either CF or PF respectively.
3. Did authors perform rotarod test on Pcp2Cre -cKO t test both motor learning and balance?
4. Supp fig 1 b needs error bars.

Responses to the reviewers

Reviewer #1:

Please note that Reviewer #1 was not able to comment on the revised manuscript, and we therefore asked one of the other reviewers on the panel to comment on your response to Reviewer #1's concerns. The reviewer agrees with Reviewer #1 that you should show the localization of endogenous DSCAM, and editorially we also agree that including this data is important to support your conclusions. Therefore, we would require that you include the smFP-tag KI mouse data in your revised manuscript.

We thank you for your important suggestions and for allowing us enough time for the experiments.

According to your recommendation, we have created the smFP-tag KI mouse line. However, unfortunately, it did not yield any appropriate signals.

Then, we tried to generate another mouse line, ALFA-tag KI, in which the ALFA-tag was designed to be knocked-in at the same region of DSCAM. This mouse line originated good signals for the endogenous DSCAM-ALFA fusion protein in the postnatal/adult cerebellum. By using this mouse line, we have successfully detected the DSCAM-ALFA fusion protein in the vicinity of several synaptic markers. This data is shown in Fig. 4e and Supplementary Fig. 2 of this re-revised manuscript.

As good results were obtained with the ALFA-tag KI mouse line, data from the HA-tag KI mouse line, used in the previous revised manuscript, were omitted in this re-revised manuscript (Fig. 1g, h, Supplementary Fig. 2 in the previous revised manuscript).

Results section, page 6, line 153-162 in revised manuscript

Next, we created a knock-in mouse line designed to express the DSCAM protein tagged with **three consecutive ALFA⁴⁹** (Fig. 1g-h, Supplementary Fig. 2a-g). **Weak but significant signals were detected in *Dscam*^{ALFA/ALFA} mice, which were hardly observed in WT**

mice (Supplementary Fig. 2e). The DSCAM-ALFA signals were preferentially detected at the Calbindin-positive regions corresponding to the dendritic structures of Purkinje cells (indicated by yellow arrowheads in Fig. 1g). Additionally, a portion of DSCAM-ALFA signals was found in the vicinity of PSD95 (postsynapse marker; yellow arrowheads in Fig. 1h), vGluT1 (PF synapse marker; white arrows in Supplementary Fig. 2f), and vGluT2 (CF synapse marker; white arrows in Supplementary Fig. 2g). These immunoblot and immunohistochemical analyses suggest, at least in part, the localisation of endogenous DSCAM at synapses on Purkinje cells.

Results section, page 10, line 291-294 in revised manuscript

Some of the immunolabellings of endogenous DSCAM-ALFA and GLAST were localised adjacent to each other at the molecular layer in the cerebellum (Fig. 4e). DSCAM-ALFA-labelled structures seem to connect with the GLAST-labelled microfibers of Bergmann glia.

Reviewer #3:

1. Reviewer appreciates including additional two co-IP experiments in supp fig 11. Unfortunately, even with both Fig 4A and supp fig 11, it is difficult to evaluate these results as DSCAM bands are too faint. Input levels for both DSCAM and Glast are also missing.

Since this is an important data reviewer suggests in addition to adding input, to perform co-IP with the known DSCAM interactors as control as well as using synaptosomes from DSCAM Pcp2 KO mice

Fig 4A and supp fig 11 please include input.

We thank the reviewer for the valuable advice and suggestion.

(i) We have added the “input” to Fig. 4A and Supplementary Fig. 11 of this re-revised manuscript.

(ii) We have generated an ALFA-tag KI mouse line to detect endogenous DSCAM (*Dscam^{ALFA}*). In this mouse line, the ALFA-tag was able to detect the localization of the endogenous DSCAM-ALFA-tag fusion protein (please see comments for the Editor). In addition, we have confirmed that the DSCAM-ALFA fusion protein was

successfully immunoprecipitated from the brain lysate of the KI mice by using an anti-ALFA-tag antibody. Then, to investigate proteins that bind to endogenous DSCAM, coimmunoprecipitation experiments were carried out using the brains of *Dscam*^{ALFA/ALFA} mice (Fig. 4b in this re-revised manuscript.) In this experiment, the endogenous DSCAM-ALFA fusion protein was coimmunoprecipitated with GLAST. As RapGEF2 is a known DSCAM interactor (Arimura et al, Sci Adv. 6(36), eaba1693, 2020), we examined RapGEF2 as a control for DSCAM-binding proteins. We successfully found that DSCAM and RapGEF2 were coimmunoprecipitated in this experiment (bottom panel of Fig. 4b in this re-revised manuscript).

As a negative control for this experiment, we performed the same coimmunoprecipitation experiment on the brains of wild-type mice (*Dscam*^{+/+}). As expected, coimmunoprecipitation was not observed for DSCAM-ALFA and GLAST or RapGEF2.

These results strongly suggest that DSCAM binds GLAST (in addition to RapGEF2) in the mouse brain. The “inputs” for these experiments were added to this figure (Fig. 4b in this re-revised manuscript).

In this coimmunoprecipitation experiment, we did not use *Pcp2-Cre; Dscam*^{flox/flox} mice as a negative control. We considered that they would not be a good negative control, as DSCAM expression should remain in interneurons and OPCs. Therefore, the ALFA-tag KI mice were used in the coimmunoprecipitation experiment, in which the WT mice (that should never express ALFA-tag) were used as a negative control.

We used the cerebellar whole lysate instead of the synaptosomal fraction in this coimmunoprecipitation experiment. If the synaptosomal fraction was to be used, at least 10 *Dscam*^{ALFA/ALFA} mice were needed. However, as this mouse strain had just been created, we did not have that many mice available. The detection sensitivity of the specific antibody against ALFA-tag was much higher than expected; therefore, the experimental results were sufficient even without the concentration of the lysate.

Results section, page 10, line 283-285 in revised manuscript

In *Dscam*^{ALFA/ALFA} mice, GLAST was found to be co-immunoprecipitated selectively with anti-

ALFA nanobody; however, this was not observed in WT mice lacking DSCAM-ALFA (Fig. 4b).

2. Because authors suggest the differential effect on DSCAM deletion on PF and CF synapses-for the fig 3 m they need to separately investigate delocalization of glast from synapses in lower (I) and higher (III) where most synapse are either CF or PF respectively.

Given the reviewer's suggestion, we attempted to distinguish between PF synapses and CF synapses in the immunoelectron microscopic data. The method was described in **Methods**, *Electron Microscopy*.

Eventually, we could obtain the data for numerous numbers of PF synapses, which was enough for the statistical analysis. However, as we observed only a small number of CF synapses, we could not perform the statistical analysis for these. Therefore, in this re-revised manuscript, we have created a new figure with the PF synapse data only (Fig. 3k-m). This suggested that GLAST molecules were significantly further away from the PSD edges in the *Dscam* mutant brain.

Methods section, page 23, line 754-757 in revised manuscript

The statistical analysis of GLAST-nanoparticle localisation was performed in PF synapses. CF and PF synapses were distinguished by morphological criteria: small terminals connecting to single spines originate from PF synapses and large terminals with multiple contacts originate from CF synapses¹¹.

3. Did authors perform rotarod test on *Pcp2Cre* -cKO t test both motor learning and balance?

No, we did not perform the rotarod test on *Dscam* mutants.
We would like to do that experiment in our future study.

4. Supp fig 1 b needs error bars.

We have added error bars in the re-revised manuscript.

REVIEWERS' COMMENTS

Reviewer #3 (Remarks to the Author):

Authors have significantly improved the manuscript by creating a novel knock-in mice and added important results showing some co-localization of DSCAM and GLAST and improved co-IP results. There are few points that should be clarified in the manuscript.

1. Authors conclude that DSCAM-GLAST interaction between the Purkinje cell dendritic spine and enwrapping Bergmann glia in PF synapses serves to localize GLAST in the peri-synaptic area of PF synapses. The key figure showing this is Fig 3. In Fig 3k authors show increased distance of PSD to Glast in PF synapses with EM. Authors-state that they distinguished PF and CF synapses in following way: "CF and PF synapses were distinguished by morphological criteria: small terminals connecting to single spines originate from PF synapses and large terminals with multiple contacts originate from CF synapses" and that "we observed only a small number of CF synapses, we could not perform the statistical analysis for these synapses"

Authors should include supplementary data showing these differences that helped them distinguish PF from CF synapses , provide explanation on why there were not enough CF synapses to quantify GLAST distance from the PSD and ideally use gold IEM with vgult1 and vglut2 to identify PF and CF synapses. Importantly, if they convincingly show this then abstract should clarify that differences in GLAST localization were seen only in PF synapses

2. In addition, EM images in Fig 3 show many dark dots that should indicate gold-labeled Glast-is this large background. With so many dots how do authors know which one is Glast?

3. In Fig. 3 authors indicate N of mice but graphs have many more dots. Please state what dots are representing (i.e. cells that were recorded from), if statistical analysis used n of mice (3 biological replicate) for number of independent samples or n of recorded cells. Also please describe why Mann Whitney U test was used and confirm that error bars are SEMs.

4. Authors state "In the presence of 100 μ M CTZ, a significant difference was detected in the amplitude ratio of PF-EPSCs (Fig. 3h-j), but not in CF-EPSCs between the WT and Dscamdel17/del17 mice (Fig. 3e-g), suggesting partial hypofunction of glutamate clearance from the PF synapses in the Dscamdel17/del17 223 cerebella. The application of CTZ tended to increase the amplitude in the early phase, but not in the late phase, of PF-EPSCs in the Dscamdel17/del17 mice (Fig. 3h)." Could authors please clarify why there was no difference in the amplitude and describe in the text what the amplitude ratio is?

5. Could authors clarify what they mean in Line 419 "In Pcp2Cre-cKO, we found the defect in motor learning (Fig. 7a-d) but in motor coordination such as during walking, grasping food, and running in daily life?"

6. In Supp Fig 10 authors state: "pre-synapses and post-synapses are tinted blue and green, respectively. Could authors please check this as it seems that PSD is present in blue areas?"

7. Could author please present VGLUT2 puncta number in Fig 6 in a manner that is consistent with the way they quantified VGLUT2 puncta in the Figs 2 and 5 showing layers I, II and III?

8. Please add the age of mice used for the co-IP experiments in Fig4 and please clarify if cerebellar or whole brain lysates were used.

Responses to the Reviewer

Authors have significantly improved the manuscript by creating a novel knock-in mice and added important results showing some co-localization of DSCAM and GLAST and improved co-IP results. There are few points that should be clarified in the manuscript.

1. Authors conclude that DSCAM-GLAST interaction between the Purkinje cell dendritic spine and enwrapping Bergmann glia in PF synapses serves to localize GLAST in the peri-synaptic area of PF synapses. The key figure showing this is Fig 3. In Fig 3k authors show increased distance of PSD to Glast in PF synapses with EM. Authors-state that they distinguished PF and CF synapses in following way: “CF and PF synapses were distinguished by morphological criteria: small terminals connecting to single spines originate from PF synapses and large terminals with multiple contacts originate from CF synapses” and that “we observed only a small number of CF synapses, we could not perform the statistical analysis for these synapses”

Authors should include supplementary data showing these differences that helped them distinguish PF from CF synapses , provide explanation on why there were not enough CF synapses to quantify GLAST distance from the PSD and ideally use gold IEM with vglut1 and vglut2 to identify PF and CF synapses. Importantly, if they convincingly show this then abstract should clarify that differences in GLAST localization were seen only in PF synapses.

We are very sorry but the previous manuscript did not cite the original important paper (textbook) below that should have been cited, and therefore the explanation of how to distinguish between these two types of synapses was not convincing.

Palay S, Chan-Palay V, Cerebellar Cortex Cytology and Organization, (1974) Cerebellar cortex.

Cytology and organization (New York: Springer).

<https://link.springer.com/book/10.1007/978-3-642-65581-4>

The textbook describes in detail how to distinguish CF synapses from PF synapses in electron microscope images, along with representative beautiful EM pictures. Many subsequent papers cite this textbook and state that CF and PF synapses are morphologically distinct. For example, the following papers.

Chen XR, Heck N, Lohof AM, Rochefort C, Morel MP, Wehrle R, Doulazmi M, Marty S, Cannaya V, Avci HX, Mariani J, Rondi-Reig L, Vodjdani G, Sherrard RM, Sotelo C, Dusart I. Mature Purkinje cells require the retinoic acid-related orphan receptor- α (ROR α) to maintain climbing fiber mono-innervation and other adult characteristics. J Neurosci. 2013 May 29;33(22):9546-62.

Cesa R, Morando L, Strata P. Glutamate receptor delta2 subunit in activity-dependent heterologous synaptic competition. *J Neurosci.* 2003 Mar 15;23(6):2363-70.

Ichikawa R, Miyazaki T, Kano M, Hashikawa T, Tatsumi H, Sakimura K, Mishina M, Inoue Y, Watanabe M. Distal extension of climbing fiber territory and multiple innervation caused by aberrant wiring to adjacent spiny branchlets in cerebellar Purkinje cells lacking glutamate receptor delta 2. *J Neurosci.* 2002 Oct 1;22(19):8487-503.

Kano M, Hashimoto K, Kurihara H, Watanabe M, Inoue Y, Aiba A, Tonegawa S. Persistent multiple climbing fiber innervation of cerebellar Purkinje cells in mice lacking mGluR1. *Neuron.* 1997 Jan;18(1):71-9.

Therefore, in this final manuscript, we have cited this textbook and added the following statement to METHODS.

Methods section (page 23, line 766-768)

CF and PF synapses were distinguished by the previously reported morphological features¹⁰². Small terminals connecting to single spines originate from PF synapses, whereas large terminals with multiple contacts originate from CF synapses.

Methods section (page 36, line 1229-1231)

102. Palay, S., Chan-Palay, V. Cerebellar Cortex Cytology and Organization. *Cerebellar cortex. Cytology and organization.* New York: Springer (1974). <https://link.springer.com/book/10.1007/978-3-642-65581-4>

According to the textbook mentioned above (Palay and Chan-Palay, 1974), the number of CF synapses per Purkinje cell in Rat is about 500, and the number of PF synapses is about 8000. In other words, in Rat, the number of CF synapses is 1% or less of all excitatory synapses on Purkinje cells. We also observed in mice that CF synapses were still less than 1~3 %. In our experimental conditions, the immunoelectron microscopy images only provided data for less than 10 synapses per image. We acquired a large number of images and analyzed them, but we still could not obtain data on a sufficient number of CF synapses to be statistically meaningful. Therefore, only PF synapses were used in this data analysis. This was described in METHODS, Electron microscopy, as below.

Methods section (page 23, line 768-772)

In our experimental conditions, the immunoelectron microscopy images only provided data for less than 10 synapses per image. Although a large number of images were acquired and analysed, a sufficient number of CF synapses was not obtained for statistical evaluation. Therefore, the statistical analysis of GLAST-nanoparticle localisation was performed only in PF synapses.

Synapses are extremely fine and dense structures. Perhaps this may be the reason why immunoelectron microscopy with secondary antibodies labelled with large gold particles of 5 nm or 10 nm did not work well with our samples and experimental conditions. Therefore, we used a secondary antibody labelled with 1.4 nm gold nanoparticles in the present study.

As 1.4 nm gold nanoparticle was very small and therefore difficult to detect directly by electron microscopy, we performed the silver enhancement after the reaction of the secondary antibody. When the silver enhancement was performed, double staining would be very difficult because the sizes of the enhanced signals were not uniform but varied. As described above, even without this double staining, the two types of synapses can be distinguished. Therefore, we performed single staining with GLAST but not double labeling with GLAST plus VGLUT1 or VGLUT2.

According to the reviewer's suggestion regarding Abstract, we revised a sentence in Abstract as below.

Abstract section (page 3, line 52-53)

These mice show impaired glutamate clearance and the delocalization of GLAST away from the cleft of parallel fibre (PF) synapse.

2. In addition, EM images in Fig 3 show many dark dots that should indicate gold-labeled Glast-is this large background. With so many dots how do authors know which one is Glast?

As mentioned above, we performed the silver enhancement to detect the 1.4 nm gold particles. Silver enhancement increases the area around the 1.4 nm particle through which the electron beam does not pass. Unlike normal gold particles of 5 nm or 10 nm, **the sizes of signals vary**, and moreover, they are not perfectly circular. In addition, the silver enhancement often causes an extremely fine-grained background. We therefore only counted signals on the plasma membrane at 10-30 nm as significant signals. We added detailed description in **METHODS** in this final manuscript.

Methods section (page 23, line 755-758)

1.4 nm gold nanoparticles (Nanogold-IgG rabbit anti-goat IgG(H+L); 2005; Nanoprobes, Yaphank, NY, USA) was incubated for 2 h, and gold nanoparticles were intensified using a silver enhancement kit (R-Gent SE-EM; Aurion, Wageningen, Netherlands).

Methods section (page 23, line 772- page 24 line 775)

Because of silver enhancement, signals varied in size and were not often round in shape. In addition, the silver enhancement caused an extremely fine-grained background. We therefore only counted signals on the plasma membrane at 10-30 nm as significant signals.

3. In Fig. 3 authors indicate N of mice but graphs have many more dots. **Please state** what dots are representing (i.e. **cells that were recorded from**), if statistical analysis used n of mice (3 biological replicate) for number of independent samples or n of recorded cells. Also please **describe** why Mann Whitney U test was used and confirm that error bars are SEMs.

The numbers of mice used were given as numbers in parentheses in the respective graphs. The dots in graphs represent cells that were recorded from. Statistical analysis was performed using data of neurons. This was described in the legend of Fig. 3.

Figure legend section (page 39, line 1321-1323 and 1327-1329)

The numbers in parentheses are mice count. Statistical analysis was performed using data from each neuron indicated by the dots in the graph.

Figure legend section (page 39-40, line 1342-1343)

Statistical analysis was performed using data from each neuron indicated by the dots in the graph.

In electrophysiological analyses using mutants or with drugs, as in the present study, the data are not normally distributed. The Mann Whitney U test can be adapted to non-normally distributed data groups and deal with relatively small sample sizes. We therefore used the Mann Whitney U test. In electrophysiological analysis experiments such as the present one, we believe it is a commonly employed statistical method.

We re-confirmed that the error bars were SEM, which is described in the figure legend.

Figure legend section (page 39, line 1321 and 1325-1326)

Data represent mean \pm SEM with individual data from each neuron

We described the reason why we used the Mann Whitney U test in METHODS, *Electrophysiology*, as below.

Methods section (page 23, line 746-749)

In electrophysiological analyses using mutants or with drugs, the data were not normally distributed. Therefore, Mann Whitney U test was used for statistical analysis, because this test can be adapted to non-normally distributed data groups and deal with relatively small sample sizes.

4. Authors state “In the presence of 100 μ M CTZ, a significant difference was detected in the amplitude ratio of PF-EPSCs (Fig. 3h–j), but not in CF-EPSCs between the WT and *Dscam*^{del17/del17} mice (Fig. 3e–g), suggesting partial hypofunction of glutamate clearance from the PF synapses in the *Dscam*^{del17/del17} 223 cerebella. The application of CTZ tended to increase the amplitude in the early phase, but not in the late phase, of PF-EPSCs in the *Dscam*^{del17/del17} mice (Fig. 3h).”
Could authors please clarify why there was no difference in the amplitude and describe in the text what the amplitude ratio is?

In this experimental system, there was a large variation in amplitude among the different stimulus locations. Therefore, we believe we could not obtain the significant difference in amplitude between WT and *Dscam*^{del17/del17}.

Then, in order to more accurately compare the contribution of Glu clearance to the AMPA response, we quantified amplitude ratios before and after CTZ administration. Taking amplitude ratios before and after CTZ administration removes the contribution of AMPAR desensitization in each response and allows a more direct assessment of the contribution of Glu clearance. We understand that this is why we were able to detect a significant increase in amplitude ratio in the *Dscam*^{del17/del17} that showed abnormal GLAST localization.

This was assessed in DISCUSSION as below.

Discussion section (page 13, line 396-402)

In this study, the application of CTZ significantly increased the amplitude ratio of PF-EPSCs in *Dscam*^{del17/del17} mice, although the amplitude was not significantly affected. In this experimental system, there was a large variation in amplitude among the different stimulus locations. That might be why we could not detect significant difference in amplitude. However, we successfully observed the significant difference by taking amplitude ratios of PF-EPSCs before and after CTZ

administration, which removed the contribution of AMPAR desensitization in each response and allowed a more direct assessment of the contribution of Glu clearance (Fig. 3e-j).

5. Could authors clarify what they mean in Line 419 “In *Pcp2Cre-cKO*, we found the defect in motor learning (Fig. 7a-d) **but in** motor coordination such as during walking, grasping food, and running in daily life?”

We were very sorry but the original description was wrong.

We corrected the sentence as below.

Results section (page 14, line 426-427)

In *Pcp2^{Cre}-cKO*, we found the defect in motor learning (Fig. 7a-d) but **not** in motor coordination such as during walking, grasping food, and running in daily life.

6. In Supp Fig 10 authors state: “pre-synapses and post-synapses are tinted blue and green, respectively.” Could authors please check this as it seems that PSD is present in blue areas?

We apologize for the confusing Fig. 10b. For clarity, we have added magnified images (Supple Fig. 10b). By the magnified images, it is clearly shown that the blue areas are pre-synapses (including synaptic vesicles) and the green areas are post-synapses (including PSD).

We added a description for the magnified images in the figure legend as below.

Supplementary Figure 10b Figure legends (Page 17)

b, Transmission electron microscopy of WT and *Dscam^{del17/del17}* mouse molecular layers. Pre-synapses (Pre) and post-synapses (Post) are tinted blue and green, respectively. **Higher magnification at the top represents the area surrounded by the white box in the larger image.** Scale bars, 500 nm.

7. Could author please present VGLUT2 puncta number in Fig 6 in a manner that is consistent with the way they quantified VGLUT2 puncta in the Figs 2 and 5 showing layers I, II and III?

In the experiment of Fig. 6, the mutant phenotype was partially but not perfectly rescued by rilusole. To be more detailed, rilusole did not well rescue the phenotype at the distal region but partially rescued at the proximal region. In addition, the degree of rescue varied very much between individual in such a drug administration experiment. In the experiment of Fig. 6, the SD of the data was relatively large in the middle region (II), and the obtained data in (II) were not suitable for

statistical analysis. Therefore, to accommodate variations in data, we analyzed the data within the relatively large framework of distal and proximal, as was presented in Fig. 6c.

8. Please add the age of mice used for the co-IP experiments in Fig4 and please clarify if cerebellar or whole brain lysates were used.

We used whole brains at P10 and P11, which was described in METHODS as below.

Methods section (page 21, line 685-686)

For the immunoprecipitation of endogenous DSCAM-ALFA, the whole brains from P10 and P11 WT (*Dscam*^{+/+}) or *Dscam*^{ALFA/ALFA} mice were suspended in RIPA buffer and sonicated for 15 min on ice.